# Learning Efficient Abstract Planning Models that Choose What to Predict

**Nishanth Kumar** *[*†]
njk@csail.mit.edu

**Willie McClinton** *[*†]
wbm3@csail.mit.edu

**Rohan Chitnis** [‡]
ronuchit@meta.com

**Tom Silver** [†]
tslvr@csail.mit.edu

**Tomás Lozano-Pérez** [†]
tlp@csail.mit.edu

**Leslie Pack Kaelbling** [†]
lpk@csail.mit.edu

[†]MIT CSAIL, [‡]Meta AI

**Abstract:** An effective approach to solving long-horizon tasks in robotics domains with continuous state and action spaces is bilevel planning, wherein a high-level search over an abstraction of an environment is used to guide low-level decision-making. Recent work has shown how to enable such bilevel planning by learning abstract models in the form of symbolic operators and neural samplers. In this work, we show that existing symbolic operator learning approaches fall short in many robotics domains where a robot's actions tend to cause a large number of irrelevant changes in the abstract state. This is primarily because they attempt to learn operators that exactly predict all observed changes in the abstract state. To overcome this issue, we propose to learn operators that 'choose what to predict' by only modelling changes *necessary* for abstract planning to achieve specified goals. Experimentally, we show that our approach learns operators that lead to efficient planning across 10 different hybrid robotics domains, including 4 from the challenging BEHAVIOR-100 benchmark, while generalizing to novel initial states, goals, and objects.

**Keywords:** Learning for TAMP, Abstraction Learning, Long-horizon Problems

## 1 Introduction

Solving long-horizon robotics problems in domains with continuous state and action spaces is extremely challenging, even when the transition function is deterministic and known. One effective strategy is to learn abstractions that capture the essential structure of the domain and then leverage hierarchical planning approaches like task and motion planning (TAMP) to solve new tasks. A typical approach is to first learn state abstractions in the form of symbolic *predicates* (classifiers on the low-level state, such as `InGripper`), then learn *operator descriptions* and *samplers* in terms of these predicates [1, 2]. The operators describe a partial transition model in the abstract space, while the samplers enable the search for realizations of abstract actions in terms of primitive actions. In this paper, we focus on the problem of learning operator descriptions from very few demonstrations given a set of predicates, an accurate low-level transition model, and a set of parameterized controllers (such as `Pick(x, y, z)`) that serve as primitive actions. We hope to leverage search-then-sample bilevel planning [3, 2, 4] using these operators to aggressively generalize to a highly variable set of problem domain sizes, initial states, and goals in challenging robotics tasks.

A natural objective for the problem of learning a good abstract model is prediction accuracy [3, 1], which would be appropriate if we were using the abstract model to make precise predictions. Instead, our objective is to find an abstract model that maximally improves the performance of the bilevel

---

*Equal contribution

7th Conference on Robot Learning (CoRL 2023), Atlanta, USA.

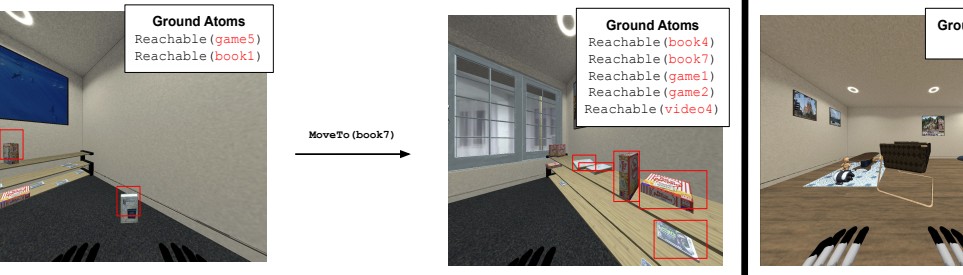

Figure 1: **Example demonstration transition and evaluation task from BEHAVIOR-100 task**. (Left) Visualization and high-level states for an example transition where the robot moves from being in range of picking a board game box and a book to picking 2 books, 2 board game boxes and a video game box. (Right) Visualization of an evaluation task where the robot starts out not in range of picking any objects.

planning algorithm, given the available data. The difference between these objectives is stark in many robotics domains where robot actions can affect many relationships among the robot and objects in the world. Making precise predictions about these state changes might require a very fine-grained abstract model with many complex operators. Such a model would require a lot of data to learn reliably, be very slow to plan with, and also be unlikely to generalize to novel tasks.

For example, consider the *Sorting Books* task from the BEHAVIOR-100 benchmark [5]. The goal is to retrieve a number of books strewn about a living room and place them on a shelf. A `Reachable(?object)` predicate is given to indicate when the robot is close enough to an object to pick it up. When the robot moves to pick a particular object, the set of objects that are reachable varies depending on their specific configuration. Figure 1 shows a transition where the robot moves to put itself in range of picking up `book7`, but happens to also be in range of picking up several other items. Optimizing prediction error on this transition would yield a very complex operator that is overfit to this specific situation and thus neither useful for efficient high-level planning nor generalizable to new tasks with different object configurations (e.g. the test situation depicted in the right panel of Figure 1, where the robot is initially not in reachable range of any objects).



**Op−MoveToBook−Prediction−Error**:
  **Args**: ?objA ?objB ?objC ?objD ?objE
       ?objF ?objG
  **Preconditions**: (**and** (Reachable ?objA)
                 (Reachable ?objB))
  **Add Effects**: (**and** (Reachable ?objC)
    (Reachable ?objD)(Reachable ?objE)
    (Reachable ?objF)(Reachable ?objG))
  **Delete Effects**: (**and** (Reachable ?objA)
                 (Reachable ?objB))





**Op−MoveToBook−Necessary−Changes**:
  **Args**: ?objA
  **Preconditions**: ()
  **Add Effects**: (Reachable ?objA)
  **Delete Effects**: ($\forall$?x. ?objA $\neq$ ?x $\Rightarrow$
    (Reachable ?x))



In this work, we take seriously the objective of learning a set of operators that optimizes overall planning performance. We observe that *operators need only model predicate changes necessary for high-level search*. Optimizing this objective enables learned operators to yield better generalization and faster planning. Our main contributions are (1) the formulation of an operator learning objective based on planning performance, (2) a procedure for distinguishing necessary changes within the high-level states of demonstrations, and (3) an algorithm that leverages (1) and (2) to learn operators via a hill-climbing search. We test our method on a wide range of complex robotic planning problems and find that our learned operators enable bilevel planning to solve challenging tasks and generalize substantially from a small number of examples.

## 2 Problem Setting

We aim to develop an efficient method for solving TAMP problems in complex domains with long-horizon solutions given structured low-level continuous state and action spaces [4, 1, 2]. We assume an 'object-oriented' state space: a state $x \in \mathcal{X}$ is characterized by the continuous proper-

ties (e.g. pose, color, material) of a set of objects. Actions, $u \in \mathcal{U}$, are short-horizon policies with both discrete and continuous parameters, which accomplish a desired change in state (e.g., Pick(block, $\theta$) where $\theta$ is a grasp transform). These *parameterized controllers* can be implemented via learning or classical approaches (e.g. motion planning). We opt for the latter in all domains in this paper. Transitions are deterministic and a simulator $f : \mathcal{X} \times \mathcal{U} \to \mathcal{X}$ predicts the next state given current state and action. The state and action representations, as well as the transition function can be acquired by engineering or learning [6, 4, 7].

PREIMAGEBACKCHAINING$((\Omega, (\overline{x}, \overline{u}), \mathcal{O}, g))$

1    $n \leftarrow \text{length}(\overline{x})$ ; $\alpha_n \leftarrow g$
2    **for** $i \leftarrow n-1, n-2, \ldots, 0$ **do**
3        $s_i, s_{i+1} \leftarrow$
          ABSTRACT$(x_i)$, ABSTRACT$(x_{i+1})$
4        $\underline{\omega}_{\text{best}} \leftarrow$ FINDBESTCONSISTENTOP$(\Omega,$
          $s_i, s_{i+1}, \alpha_{i+1}, \mathcal{O})$
5        **if** $\underline{\omega}_{best} = Null$ **then**
          |   **break**
6        $\underline{\omega}_{i+1} \leftarrow \underline{\omega}_{\text{best}}$
7        $\alpha_i \leftarrow \underline{P_{i+1}} \cup (\alpha_{i+1} \setminus \underline{E_{i+1}^+})$
8    **return** $(\underline{\omega}_{i+1}, \ldots, \underline{\omega}_n), (\alpha_i, \ldots, \alpha_n)$

**Algorithm 1:** Preimage backchaining procedure (details in (§C)).

HILLCLIMBINGSEARCH$(\mathcal{D})$

1    $J_{\text{last}} \leftarrow \infty$; $J_{\text{curr}} \leftarrow \text{J}(\Omega, \mathcal{D})$
2    $\Omega \leftarrow \emptyset$
3    **while** $J_{curr} < J_{last}$ **do**
4        $\Omega' \leftarrow$ IMPROVECOVERAGE$(\Omega, \mathcal{D})$
5        **if** $\text{J}(\Omega', \mathcal{D}) < J_{curr}$ **then**
6        |   $\Omega \leftarrow \Omega'$ ; $J_{\text{curr}} \leftarrow \text{J}(\Omega, \mathcal{D})$
7        $\Omega' \leftarrow$ REDUCECOMPLEXITY$(\Omega, \mathcal{D})$
8        **if** $\text{J}(\Omega', \mathcal{D}) < J_{curr}$ **then**
9        |   $\Omega \leftarrow \Omega'$ ; $J_{\text{curr}} \leftarrow \text{J}(\Omega, \mathcal{D})$
10       $J_{\text{last}} \leftarrow J_{\text{curr}}$ ; $J_{\text{curr}} \leftarrow \text{J}(\Omega, \mathcal{D})$
11    **return** $\Omega$

**Algorithm 2:** HILLCLIMBINGSEARCH learns operators that optimize objective $J$.

Since planning in this low-level space can be expensive and unreliable [8, 1, 2], we pursue a search-then-sample bilevel planning strategy in which search in an abstraction is used to guide low-level planning. We assume we are given a set of *predicates* $\Psi$ to define discrete properties of and relations between objects (e.g., On) via a classifier function that outputs true or false for a tuple of objects in a low-level state. Predicates induce a state abstraction ABSTRACT $: \mathcal{X} \to \mathcal{S}$ where ABSTRACT$(x)$ is the set of true *ground atoms* in $x$ (e.g., {HandEmpty(robot), On(b1, b2), ...}). The low-level state space, action space, and simulator together with the predicates comprise an *environment*. To enable bilevel planning for an environment, we must *learn* a partial abstract transition model over the predicates in the form of symbolic operators.

We consider a standard learning-from-demonstration setting where we are given a set of training tasks with demonstrations, and must generalize to some held-out test tasks. A *task* $T \in \mathcal{T}$ is characterized by a set of objects $\mathcal{O}$, an initial state $x_0 \in \mathcal{X}$, and a goal $g$. The goal is a set of ground atoms and is *achieved* in $x$ if $g \subseteq$ ABSTRACT$(x)$. A *solution* to a task is a sequence of actions $\overline{u} = (u_1, \ldots, u_n)$ that achieve the goal ($g \subseteq$ ABSTRACT$(x_n)$, and $x_i = f(x_{i-1}, u_i)$ for $1 \le i \le n$) from the initial state. Each environment is associated with a *task distribution*. Our objective is to maximize the likelihood of solving tasks from this distribution within a planning time budget.

## 3   Operators for Bilevel Planning

Symbolic operators, defined in PDDL [9] to support efficient planning, specify a transition function over our state abstraction. Formally, an operator $\omega$ has *arguments* $\overline{v}$, *preconditions* $P$, *add effects* $E^+$, *delete effects* $E^-$, and a *controller* $C$. The preconditions and effects are each expressions over the arguments that describe conditions under which the operator can be executed (e.g. Reachable(?book1)) and resulting changes to the abstract state (e.g. Holding(?book1)) respectively. The controller is a policy (from the environment's action space), parameterized by some of the discrete operator arguments, as well as continuous parameters whose values will be chosen during the sampling phase of bilevel planning. A substitution of arguments to objects induces a *ground operator* $\underline{\omega} = \langle \underline{P}, \underline{E^+}, \underline{E^-}, \underline{C} \rangle$. Given a ground operator $\underline{\omega} = \langle \underline{P}, \underline{E^+}, \underline{E^-}, \underline{C} \rangle$, if $\underline{P} \subseteq s$, then the successor abstract state $s'$ is $(s \setminus \underline{E^-}) \cup \underline{E^+}$ where $\setminus$ is set difference. We use $F(s, \underline{\omega}) = s'$ to denote this (partial) abstract transition function.

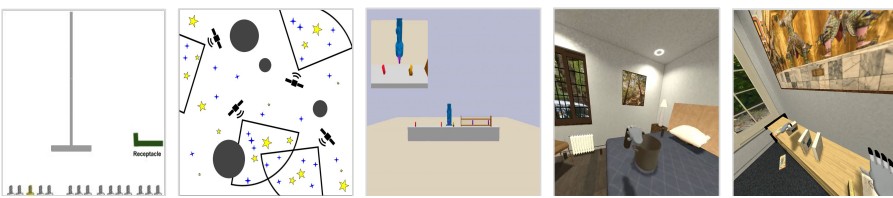

Figure 2: **Environments**. Visualizations for *Screws*, *Satellites*, *Painting*, *Collecting Cans*, and *Sorting Books*.

In previous work on operator learning for bilevel planning [3, 2], operators are connected to the underlying environment via the following semantics: if $F(s, \underline{\omega}) = s'$, then *there exists* some low-level transition $(x, u, x')$ where $\text{ABSTRACT}(x) = s$, $\text{ABSTRACT}(x') = s'$, and $u = \underline{C}(\theta)$ for some $\theta$. These semantics embody the "prediction error" view in that $\text{ABSTRACT}$ must predict the entire next state ($\text{ABSTRACT}(x') = s'$). Towards implementing the alternative "necessary changes" view, we will instead only require the abstract state output by $F$ to be a *subset* of the atoms in the next state, that is, $\text{ABSTRACT}(x') \subseteq s'$. Thus, we permit our operators to have universally quantified single-predicate delete effects (e.g., $\forall?\text{v}. \quad \text{Reachable}(?\text{v})$). Intuitively, $F$ is now responsible only for predicting abstract *subgoals* to guide low-level planning, rather than predicting entire successor abstract states. Under these new semantics, $F$ encodes an abstract transition model where the output atoms $s'$ represent the set of possible abstract states where these atoms hold.

Given operators, we can solve new tasks via search-then-sample bilevel planning (decribed in detail in appendix (§A) and previous work [10, 2, 3, 4, 1]). Given the task goal $g$, initial state $x_0$, and corresponding abstract state $s_0 = \text{ABSTRACT}(x_0)$, bilevel planning uses AI planning techniques (e.g., Helmert [11]) to generate candidate *abstract plans*. An abstract plan is a sequence of ground operators $(\underline{\omega}_1, \dots, \underline{\omega}_n)$ where $g \subseteq s_n$ and $s_i = F(s_{i-1}, \underline{\omega}_i)$ for $1 \le i \le n$. The corresponding abstract state sequence $(s_1, \dots, s_n)$ serves as a sequence of *subgoals* for low-level planning. The controller sequence $(\underline{C}_1, \dots, \underline{C}_n)$ provides a *plan sketch*, where all that remains is to *refine* the plan by "filling in" the continuous parameters. We sample continuous parameters $\theta$ for each controller starting from the first and checking if the controller achieves its corresponding subgoal $s_i$. If we cannot sample such parameters within a constant time budget, we backtrack and resample, and eventually even generate a new abstract plan. We adapt previous work [2] to learn neural network samplers after learning operators (see (§D) in the appendix for more details).

## 4    Learning Operators from Demonstrations

To enable efficient bilevel planning, operators must generate abstract plans corresponding to plan sketches that are likely to be refinable into low-level controller executions. Given demonstrations $\mathcal{D}$ consisting of a goal $g$, action sequence $\overline{u}$, corresponding state sequence $\overline{x}$, and set of objects $\mathcal{O}$, we wish to find the simplest set of operators such that for every training task, abstract planning with these operators is able to generate the plan sketch corresponding to the demonstration for this task.

Specifically, we minimize the following objective:

$$J(\Omega) \triangleq (1 - \texttt{coverage}(\mathcal{D}, \Omega)) + \lambda \texttt{complexity}(\Omega) \tag{1}$$

where $\texttt{coverage}$ is the fraction of demonstration plan sketches that planning with operator set $\Omega$ is able to produce and $\texttt{complexity}$ is the number of operators. In our experiments, we set $\lambda$ to be small enough so that complexity is never decreased at the expense of coverage.

We will approach this optimization problem using a hill-climbing search (Algorithm 2), which benefits from having a more fine-grained interpretation of coverage defined over *transitions* within a trajectory instead of entire trajectories. To develop this definition, we will first introduce the notion of *necessary atoms*.

**Definition 4.1.** Given an operator set $\Omega$, and an abstract plan $(\underline{\omega}_1, \dots, \underline{\omega}_n)$ in terms of $\Omega$ that achieves goal $g$, the *necessary atoms* at step $n$ are $\alpha_n \triangleq g$, and at step $0 \le i < n$ are $\alpha_i \triangleq \underline{P_{i+1}} \cup (\alpha_{i+1} \setminus \underline{E_{i+1}^+})$.

In other words, an atom is necessary if it is mentioned either in the goal, or the preconditions of a future operator in a plan. If operators in $\Omega$ model each of the necessary atoms for every timestep, then these operators must be sufficient for producing the corresponding abstract plan via symbolic planning. Moreover, each necessary atom set is minimal in that no atoms can be removed without violating either the goal or a future operator's preconditions. Thus, we need only learn operators to model changes in these necessary atoms.

Given necessary atoms, we can now define what it means for some sequence of operators to be *consistent* with some part of a demonstration.

**Definition 4.2.** An abstract plan suffix $(\underline{\omega}_k, \ldots, \underline{\omega}_n)$ using operators from set $\Omega$ and with necessary atoms $(\alpha_{k-1}, \ldots, \alpha_n)$ is *consistent* with a demonstration $(\overline{x}, \overline{u})$ for goal $g$ and timesteps $1 \leq k \leq n$, where $\overline{x} = (x_0, \ldots, x_n)$ and $\overline{u} = (u_1, \ldots, u_n)$ if for $k \leq i \leq n$, (1) the states are consistent: $\alpha_i \subseteq F(\text{ABSTRACT}(x_{i-1}), \underline{\omega}_i) \subseteq \text{ABSTRACT}(x_i)$; and (2) the actions are consistent: if the controller for $\underline{\omega}_i$ is $\underline{C}_i$, then $u_i = \underline{C}_i(\theta)$ for some $\theta$.

Note that consistency is defined at the level of individual transitions within a demonstration. Thus, it is useful even if $\Omega$ does not contain sufficient operators to produce a plan that is consistent with an entire demonstration. Given this notion, we can now define *coverage* for a particular demonstration.

**Definition 4.3.** The *demonstration coverage*, $\eta(\Omega, (\overline{x}, \overline{u}, g, \mathcal{O}))$ of demonstration $(\overline{x}, \overline{u})$ of length $n$ for goal $g$ and objects $\mathcal{O}$ by operators $\Omega$, is a number from 0 to $n$ indicating the length of the longest consistent suffix for the demonstration that can be generated with $\Omega$ ground with objects $\mathcal{O}$.

Now, we define a fine-grained notion of coverage for optimization via hill-climbing:

$$\texttt{coverage}(\Omega, \mathcal{D}) = \Sigma_{(\overline{x}, \overline{u}, g, \mathcal{O}) \in \mathcal{D}} \frac{\eta(\omega, (\overline{x}, \overline{u}, g, \mathcal{O}))}{|\overline{u}|} \tag{2}$$

Computing `coverage` implicitly requires that we compute an abstract plan suffix, as well as necessary atoms (in order to compute $\eta$). To do this efficiently, we will leverage the fact that we know the necessary atoms for the final timestep of every demonstration (they are simply the goal atoms!), and compute necessary atoms and plan suffixes for previous timesteps via a *preimage backchaining* procedure that starts at the goal and works backwards given operators in $\Omega$. Pseudocode for this algorithm is shown in Algorithm 1. We pass backward through each transition in a demonstration trajectory while attempting to choose an operator in $\Omega$ to cover (per Definition 4.2) the transition and then updating the transition's necessary atoms using Definition 4.1. If there are multiple operators in $\Omega$ that all satisfy the conditions of Definition 4.2, we use a heuristic that selects the operator that best matches this particular transition (see (§C) in the appendix for details).

**Hill-Climbing Search.** We perform a hill-climbing search on $J(\Omega)$ (Equation 1), starting with the empty operator set. We have two search-step generators: the IMPROVECOVERAGE generator picks a demonstration that is not completely covered by the current operator set and proposes a change that will increase the `coverage` term of the objective, and the REDUCECOMPLEXITY generator simply proposes to delete operators to decrease the `complexity` term.

**IMPROVECOVERAGE generator.** Given a candidate operator set $\Omega$, preimage backchaining is used to identify an abstract plan suffix of length $k$ with corresponding necessary atoms $(\alpha_{k-1}, \ldots, \alpha_n)$ that is consistent with a demonstration $(\overline{x}, \overline{u})$ for goal $g$, where $\overline{x} = (x_0, \ldots, x_n)$ and $\overline{u} = (u_1, \ldots, u_n)$, and where $1 \leq k < n$. Since $k < n$, we know the transition $(x_{k-2}, u_{k-1}, x_{k-1})$ with necessary atoms $\alpha_{k-1}$ is not covered by any operator in $\Omega$. To improve the `coverage` term in our objective, we wish to generate at least one new operator $\omega$ to cover this transition without uncovering any others. Definition 4.2 gives us the following constraints on the new operator $\omega$ with grounding $\underline{\omega}$:

1. The controller $\underline{C}$ must match $u_{k-1}$.
2. $\underline{P} \subseteq s_{k-2}$, i.e., the preconditions must be satisfied in the previous state.
3. $((s_{k-2} \setminus s_{k-1}) \cap \alpha_{k-1}) \subseteq \underline{E}^+$, i.e., the add effects must include the added necessary atoms.
4. $F(s_{k-2}, \underline{\omega}) \subseteq s_{k-1}$, i.e., the delete effects should remove all atoms in $s_{k-2}$ but not in $s_{k-1}$.
5. $\overline{v}$ must at least include one variable (with corresponding type) for each of the objects in 1–4.

| Environment | Ours | LOFT | LOFT+Replay | CI | CI + QE | GNN Shoot | GNN MF |
|---|---|---|---|---|---|---|---|
| Painting | **98.80 (0.42)** | 0.00 (0.00) | **98.20 (0.91)** | **99.00 (0.31)** | 93.40 (1.47) | 36.00 (3.39) | 0.60 (0.29) |
| Satellites | **93.40 (3.52)** | 0.00 (0.00) | 34 (5.28) | **91.60 (2.68)** | **95.20 (1.30)** | 40.40 (3.04) | 11.00 (1.44) |
| Cluttered 1D | **100.00 (0.00)** | 17.20 (5.46) | 0.00 (0.00) | 17.40 (5.52) | 92.80 (0.90) | **98.60 (0.63)** | **98.60 (0.63)** |
| Screws | **100.00 (0.00)** | 0.00 (0.00) | 0.00 (0.00) | 0.00 (0.00) | 50.00 (15.81) | 95.60 (3.05) | 95.80 (3.07) |
| Cluttered Satellites | **95.20 (0.75)** | 0.00 (0.00) | 0.00 (0.00) | 1.60 (0.61) | 6.00 (1.57) | 4.80 (1.27) | 0.00 (0.00) |
| Cluttered Painting | **99.20 (0.42)** | 0.00 (0.00) | 0.00 (0.00) | 0.00 (0.00) | 0.00 (0.00) | 4.60 (1.16) | 0.00 (0.00) |
| Opening Presents | **100.00 (0.00)** | 0.00 (0.00) | - | 83.00 (10.77) | 83.00 (10.77) | 28.00 (5.96) | 0.00 (0.00) |
| Locking Windows | **100.00 (0.00)** | 0.00 (0.00) | - | 90.00 (4.47) | 88.00 (4.42) | 0.00 (0.00) | 0.00 (0.00) |
| Collecting Cans | **77.00 (11.75)** | 0.00 (0.00) | - | 0.00 (0.00) | 1.00 (0.94) | 0.00 (0.00) | 0.00 (0.00) |
| Sorting Books | **69.00 (11.61)** | 0.00 (0.00) | - | 0.00 (0.00) | 0.00 (0.00) | 0.00 (0.00) | 0.00 (0.00) |

Table 1: Percentage success rate on test tasks for all domains. Note that BEHAVIOR domains use training and testing sizes of 10 tasks, while all other domains use 50 tasks. The percentage standard error is shown in brackets. All entries within one standard error from the best mean success rate are bolded.

These constraints exactly determine $\omega$'s controller, but there are many possible choices for the other components that would satisfy 2–5. One straightforward choice would be to optimize prediction error over the single transition $(s_{k-2}, u_{k-1}, s_{k-1})$ and necessary atoms $\alpha_{k-1}$, but this would lead to a hyper-specific operator.

Instead, we construct a more general operator that covers this transition as well as other transitions in the data set. Specifically, we set $\omega$'s ground controller $\underline{C} = u_{k-1}$ to satisfy condition 1 above, and ground add effects $\underline{E}^+ = ((s_{k-2} \setminus s_{k-1}) \cap \alpha_{k-1})$ to satisfy condition 3. We then construct the controller $C$ and add effects $E^+$ by 'lifting': i.e, replacing all objects that appear in $\underline{C}$ and $\underline{E}^+$ by variables of the same type. We record the substitution necessary between the variables and objects for this transition as $\delta_\tau$. The operator's arguments are then the union of the variables that appear here. To find preconditions and delete effects that satisfy conditions 2 and 4, we will use each substitution $\delta_\tau$ associated with each transition in $\mathcal{D}_\omega$ to *lift* each transition $\tau$, replacing all objects with arguments while discarding any atom with objects not in $\delta_\tau$. Following Chitnis et al. [2], the *preconditions* are then: $P \leftarrow \bigcap_{\tau=(s_i,\cdot,\cdot)\in\mathcal{D}_\omega} \delta_\tau(s_i)$. Similarly, the *atomic delete effects* are: $E_\circ^- \leftarrow \bigcup_{\tau=(s_i,\cdot,s_{i+1})\in\mathcal{D}_\omega} \delta_\tau(s_{i+1}) \setminus \delta_\tau(s_i)$. Finally, the operator's *quantified delete effects* are set to be the set of predicates that appear in the current predicted abstract state, but not the observed abstract state: $F(s_i, \underline{\omega}) \setminus s_{i-1}$. The final delete effects are the union of the atomic and quantified.

Some additional bookkeeping is necessary once this new operator has been induced to guarantee that this generator decreases the coverage term (we must recompute preconditions and delete effects for *all* current candidate operators, since their respective transition datasets might have changed, etc.), and we detail these, along with a proof of termination, in the appendix (§B.1).

**REDUCECOMPLEXITY generator.** Given a candidate operator set $\Omega$, we simply delete a single operator from the set and then recompute preconditions and delete effects for all remaining operators as described above (since now, their associated transition datasets $\mathcal{D}_\omega$ might change).

## 5 Experimental Results

Our experiments are designed to empirically answer the following questions:

- **Q1.** Does our approach learn operators capable of generalizing to novel goals and situations with different objects than seen during training?
- **Q2.** How effective is bilevel planning using our learned operators vs. existing baselines?
- **Q3.** How does the performance of our approach vary with the amount of training data?

For additional analyses, including learning efficiency and scalability, complexity of learned operators, task planning efficiency, and ablations of our approach, see (§G) and (§H) in the appendix.

**Environments.** We provide high-level environment details, with specifics in the appendix (§E). We deliberately include a few simple environments to highlight differences between methods.

- *Painting*: a challenging robotics environment used by Silver et al. [3, 1]. A robot in 3D must pick, wash, dry, paint, and then place various objects.
- *Satellites*: a 2D environment inspired by the Satellites domain from Bacchus [12], but augmented with collisions and realistic sensing constraints. See appedix (§E) for further details.

- *Cluttered 1D*: a simple environment where the robot must move and collect objects cluttered along a 1D line. An object can only be collected if it is reachable.
- *Screws*: a 2D environment where the agent controls a magnetic crane and must pick specific screws from clutter to place them into a receptacle.
- *Cluttered Satellites*: same as "Satellites", except readings must be taken from multiple objects.
- *Cluttered Painting*: same as Painting, except the robot can be next to many objects at a time.
- *BEHAVIOR-100 Tasks*: a set of complex, long-horizon household robotic tasks simulated with realistic 3D models of objects and homes [5]. In *Opening Presents*, the robot must open a number of boxes. In *Locking Windows*, the robot must close a number of open windows. In *Collecting Cans*, the robot must pick up a number of empty soda cans strewn amongst the house and throw them into a trash can. In *Sorting Books*, the robot must find books in a living room and place them each onto a cluttered shelf.

**Approaches.** We describe the approaches we compare against, with details in the appendix (§F).

- *Cluster and Intersect (CI)* [1]: Induces a different operator for every set of unique lifted effects.
- *LOFT* [3]: Optimizes prediction error, but with a more general class of preconditions. We include a version that only learns from demonstrations (LOFT) and a version that collects additional transitions (2500) in each domain (LOFT+Replay) as in the original work. Collecting replay data in BEHAVIOR was intractable due to the size and complexity of the simulation.
- *Cluster and Intersect with Quantified Delete Effects (CI + QE)*: A variant of Cluster and Intersect that is capable of learning operators that have both atomic and quantified delete effects. It first runs Cluster and Intersect, and then induces quantified delete effects by optimizing prediction error via a hill-climbing search.
- *GNN Shooting*: Trains an abstract GNN policy with behavioral cloning and uses it for trajectory optimization. This is inspired by a baseline from [1], see (§F) in the appendix for details.
- *GNN Model-Free*: Uses the same trained GNN as above, but directly executes the policy.

**Experimental Setup.** For the non-BEHAVIOR environments, we run all methods on up to 50 training demonstrations. For BEHAVIOR environments, we use 10, since collecting training data in these complex environments is very time and memory intensive. Training demonstrations were collected by a hand-coded 'oracle' policy that was specific to each environment. All experiments were conducted on a quad-core Intel Xeon Platinum 8260 processor with a 192GB RAM limit, and all results are averaged over 10 random seeds. For each seed, we sample a set of *evaluation tasks* from the task distribution $\mathcal{T}$. *The evaluation tasks have more objects, different initial states, and more atoms in the goal than seen during training.* Our key measure of effective bilevel planning is success rate within a timeout (10 seconds for non-BEHAVIOR environments, 500 seconds for 3 BEHAVIOR environments, and 1500 seconds for "Sorting Books": the most complex environment).

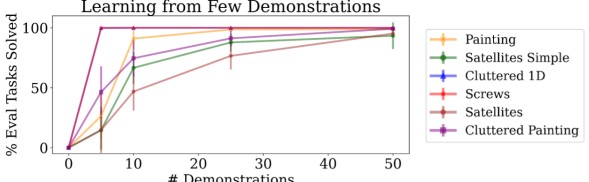

Figure 3: Data-efficiency of main approach.

**Results and Analysis.** Table 1 shows the success rate achieved on testing tasks for all our environments and methods. Our method solves many more held-out tasks within the timeout than the baselines, demonstrating its efficacy even in environments where an assumption on downward refinablity might not hold. In our simpler environments (Painting and Satellites; Opening Presents and Locking Windows for BEHAVIOR), the controllers cause a small number of changes in the abstract state, and baseline approaches (CI, CI + QE) perform reasonably well. In all the other environments, the controllers cause a large number of changes in the abstract state, and the performance of operator learning baselines degrades substantially, though GNN-baselines perform well on Cluttered 1D

and Screws. Despite the increased complexity, our approach learns operators that are resilient to the lack of downward refinability, enabling bilevel planning to achieve a substantial test-time success rate under timeout. The performance in Collecting Cans and Sorting Books is especially notable; all baselines achieve a negligible success rate, while our approach achieves a near 70% rate on testing tasks. Upon investigation, we found that failures are due to local minima during learning for certain random seeds. Figure 3 shows our method's testing success rate as a function of the size of its training set for our non-BEHAVIOR environments. Our approach's performance improves with more data, though as the dataset size increases, the impact of additional data on performance reduces.

## 6  Related Work

Our work continues a long line of research in learning operators for planning [13, 14, 15, 16, 17, 18, 19, 20, 21]; see Arora et al. [22] for a recent survey. This previous work focuses on learning operators from purely discrete plan traces in the context of classical (not bilevel) planning.

Other work has considered learning symbolic planning models in continuous environments [23, 24, 25, 26, 27, 28, 29, 30, 31], but typically the interface between symbolic planner and low-level policies assumes downward refinability, which requires that every valid high-level plan must be refinable into low-level steps [32], a critical assumption we do not make. Therefore, our efforts are most directly inspired by LOFT [3] and learning Neuro-Symbolic Relational Transition Models [2], which optimize prediction error to learn operators for bilevel planning. Like our method, LOFT performs a search over operator sets, but commits to modeling all effects seen in the data and searches only over operator preconditions. We point out the limitations of optimizing prediction error in complex robotics environments, and take inspiration from Silver et al. [1] who show that optimizing for planning efficiency can enable good predicate invention. We include LOFT and Cluster and Intersect (used by [2, 1]) as baselines representative of these previous methods in our experiments.

The bilevel planner used in our work can be viewed as a search-then-sample solver for (TAMP) [33, 34, 10]. This bilevel strategy allows for fast planning in continuous state and action spaces, while avoiding the downward refinability assumption. To that end, our work also contributes to a recent line of work on learning for TAMP. Other efforts in this line include sampler learning [35, 36, 37, 38], heuristic learning [39, 40, 41], and abstract plan feasibility estimation [42, 43].

## 7  Limitations, Conclusions and Future Work

Our method assumes that the provided predicates $\Psi$ comprise a good state abstraction given the task distribution for operator learning. With random or meaningless predicates, our approach is likely to learn complex operators such that planning with these is unlikely to outperform non-symbolic behavior-cloning baselines. Fortunately, prior work [1] suggests such 'good' predicates can be learned from data. There is no guarantee that our overall hill-climbing procedure will converge quickly; the IMPROVECOVERAGE successor generator is especially inefficient and has a high worst-case computational complexity. In practice, we find its learning time to be comparable or faster than baseline methods in our domains (see (§G) in the appendix), though this may not hold in more complex domains where a very large (e.g. greater than 100) number of operators need to be learned. Additionally, our IMPROVECOVERAGE successor generator is rather complicated, and there is perhaps a simpler and more efficient way to optimize the `coverage` term in our objective.

Overall, we proposed a new objective for operator learning that is specifically tailored to bilevel planning, and a search-based method for optimizing this objective. Experiments confirmed that operators learned with our new method lead to substantially better generalization and planning than those learned by optimizing prediction error and other reasonable baselines. Important next steps include learning all components necessary to enable bilevel planning, including the predicates [1], controllers [4], and object-oriented state-space, as well as handling stochasticity and partial observability. We believe that pursuing these steps will yield important progress toward solving sparse-feedback, long-horizon decision-making problems at scale via modular learning.

## Acknowledgements

We gratefully acknowledge support from NSF grant 2214177; from AFOSR grant FA9550-22-1-0249; from ONR MURI grant N00014-22-1-2740; from ARO grant W911NF-23-1-0034; from the MIT-IBM Watson Lab; from the MIT Quest for Intelligence; and from the Boston Dynamics Artificial Intelligence Institute. Nishanth, Willie, and Tom are supported by NSF Graduate Research Fellowships. Any opinions, findings, and conclusions or recommendations expressed in this material are those of the authors and do not necessarily reflect the views of our sponsors. We thank Jorge Mendez, Aidan Curtis, and anonymous conference reviewers for helpful comments on earlier drafts of this work.

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

## A  Bilevel Planning Algorithm Details

Algorithms 3 and 4 provide pseudocode and additional details necessary for high-level and low-level search respectively in bilevel planning (Algorithm 5).

GENABSTRACTPLAN($s_0$, $g$, $\Omega$, $n_{abstract}$)

1    $\underline{\Omega} \leftarrow$ GROUND($\Omega$)

     // Search over ground operators from s0 to goal (returns top n plans).

2    $\hat{\pi} \leftarrow$ SEARCH($s_0, g, \underline{\Omega}, n_{\text{abstract}}$)

3    **return** $\hat{\pi}$

**Algorithm 3:** This is GENABSTRACTPLAN which finds a high-level plan by creating operators for all possible groundings then uses search to find $n_{\text{abstract}}$ plans. It returns a list of plans $\hat{\pi}$, which the REFINE procedure below will attempt to turn into executable trajectories.

REFINE(($\hat{\pi}$, $x_0$, $\Psi$, $\Sigma$, $n_{samples}$))

1    state $\leftarrow$ ABSTRACT($x_0$)

     // While current state is not goal, sample and run current operator on current state and check ground atoms.  If is passes continue if not backtrack.

2    $\text{curr}_{\text{idx}} \leftarrow 0$

     **while** $\text{curr}_{\text{idx}} < len(\hat{\pi})$ **do**

3        $\text{samples}[\text{curr}_{\text{idx}}] \leftarrow samples[curr_{idx}] + 1$

4        $\text{state}_{\text{current}}, \underline{\Omega} \leftarrow \hat{\pi}[\text{curr}_{\text{idx}}]$

5        $\underline{\Omega}.C.\Theta \sim \underline{\Omega}.\Sigma$

6        $\pi[\text{curr}_{\text{idx}}] \leftarrow \underline{\Omega}.C$

7        $\text{curr}_{\text{idx}} \leftarrow \text{curr}_{\text{idx}} + 1$

       **if** $\underline{\Omega}.C.initiable(state_{current})$ **then**

8          $\text{state}_{\text{next}} \leftarrow$ Simulate($\text{state}_{\text{current}}, \underline{\Omega}.C$)

9          $\text{state}_{\text{expected}}, \_ \leftarrow \hat{\pi}[\text{curr}_{\text{idx}}]$

         **if** $state_{next} \subseteq state_{expected}$ **then**

10            can_continue_on $\leftarrow True$

           **if** $curr_{idx} == len(skeleton)$ **then**

11             **return** sucess, $\pi$

         **else**

12            canContinueOn $\leftarrow False$

       **else**

13          canContinueOn $\leftarrow False$

       **if** *not canContinueOn* **then**

14          $\text{curr}_{\text{idx}} \leftarrow \text{curr}_{\text{idx}} - 1$

         **if** $samples[curr_{idx}] == max\_samples$ **then**

15            **return** failure, $\pi$

16    **return** success, $\pi$

**Algorithm 4:** This is REFINE which turns a task plan $\hat{\pi}$ into a sequence of ground skills. It gets the state and operators from $\hat{\pi}$ and adds the controller with newly sampled continuous parameters to $\pi$. After this it checks to see if the added controller is initiable from the current state in the plan and we simulate the skill execution to verify it reached the expected state we predicted next in $\hat{\pi}$. If the controller is not initiable or fails the expected atoms check we backtrack and resample a new continuous parameter for this controller until either we reach the max number of samples or we successfully refine our final controller. In this way, the performance of the low-level controllers in the environment strongly impacts the overall planning process.

BILEVELPLANNING($\mathcal{O}$, $x_0$, $g$, $\Psi$, $\Omega$, $\Sigma$)

```
      // Parameters:  n_abstract, n_samples.
1     s_0 ← ABSTRACT(x_0)
      // Outer Planning Loop
2     for π̂ in GENABSPLAN(s_0, g, Ω, n_abstract) do
        // Inner Refinement Loop
3       if REFINE(π̂, x_0, Ψ, Ω, n_samples) succeeds w/ π return π
```

**Algorithm 5:** Pseudocode for bilevel planning algorithm, adapted from Silver et al. [1]. The inputs are objects $\mathcal{O}$, initial state $x_0$, goal $g$, predicates $\Psi$, operators $\Omega$, and samplers $\Sigma$; the output is a plan $\pi$. The outer loop GENABSPLAN generates high-level plans that guide our inner loop, which samples continuous parameters from our samplers $\Sigma$ to concretize each abstract plan $\hat{\pi}$ into a plan $\pi$. If the inner loop succeeds, then the found plan $\pi$ is returned as the solution; if it fails, then the outer GENABSTRACTPLAN continues.

IMPROVECOVERAGE($\Omega$, $\mathcal{D}$)

```
1     cov_init, D_α, τ_unc, α_unc ← COMPUTECOVERAGE(Ω, D)
2     if cov_init = |D| then
        return Ω
3     cov_curr ← cov_init
4     Ω' ← Ω
5     while cov_curr ≥ cov_init do
6       ω_new ← INDUCEOPTOCOVER(τ_unc, α_unc)
7       Ω' ← REMOVEPRECANDDELEFFS(Ω') ∪ ω_new
8       (D_ω1 ... D_ωm), (δ_τ1, ..., δ_τj) ← PARTITIONDATA(Ω', D)
9       Ω' ← INDUCEPRECANDDELEFFS(Ω',
          (D_ω1 ... D_ωm),
          (δ_τ1, ..., δ_τj))
10      Ω' ← Ω' ∪ ENSURENECATOMSSAT(ω_new, D_α)
11      (D_ω1 ... D_ωl), (δ_τ1, ..., δ_τj) ← PARTITIONDATA(Ω', D)
12      Ω' ← INDUCEPRECANDDELEFFS(Ω',
          (D_ω1 ... D_ωl),
          (δ_τ1, ..., δ_τj))
13      cov_curr, D_α, τ_unc, α_unc ← COMPUTECOVERAGE(Ω', D)
14    Ω' ← PRUNENULLDATAOPERATORS(Ω')
15    return Ω'
```

**Algorithm 6:** Pseudocode for our `improve-coverage` successor generator. The inputs are a set of operators $\Omega$, the set of all training demonstrations $\mathcal{D}$, and the corresponding set of training tasks $\mathcal{T}_{\text{train}}$. The output is a set of operators $\Omega'$ such that `coverage`($\Omega'$) $\leq$ `coverage`($\Omega$).

# B    Detailed Description of Successor Generators

## B.1    IMPROVECOVERAGE

The pseudocode for the `improve-coverage` successor generator is shown in Algorithm 6. Given the current candidate operator set $\Omega$, training demonstrations $\mathcal{D}$ and corresponding tasks $\mathcal{T}_{\text{train}}$, we first attempt to compute the current coverage of $\Omega$ on $\mathcal{D}$. We do this by calling the COMPUTE-COVERAGE method. This method simply calls Algorithm 1 on every demonstration $(\overline{x}, \overline{u})$ in $\mathcal{D}$ (the set of objects $\mathcal{O}$ and goal $g$ required by Algorithm 1 are obtained from the training tasks). The COMPUTECOVERAGE method then returns the number of covered transitions[2] ($\text{cov}_{\text{init}}$), a dataset of necessary atoms sequences for each demonstration Algorithm 1 is able to cover ($\mathcal{D}_\alpha$), the first uncovered transition encountered ($\tau_{\text{unc}} = (s_k, u_{k+1}, s_{k+1})$), and the corresponding necessary atoms for the transition ($\alpha_{\text{unc}}$). If the number of covered transitions is the same as the size of the training dataset, then all transitions must be covered and the `coverage` term in our objective (Equation

---

[2]The total number of transitions in abstract plan suffixes that Algorithm 1 is able to find when run on each demonstration in $\mathcal{D}$.

1) must be 0. We thus just return the current operator set $\Omega$ with no modifications. Otherwise, we compute a new set of operators $\Omega'$ with a lower `coverage` value.

To generate $\Omega'$, we first create a new operator with preconditions, add effects and arguments set to cover the transition $\tau_{\text{unc}}$ and corresponding necessary atoms $\alpha_{\text{unc}}$. The operator's ground controller $\underline{C}(\theta) = u_{k+1}$ is determined directly from the transition's action $u_{k+1}$. The operator's ground add effects are set to be $\underline{E}^+ = (s_{k+1} \setminus s_k) \cap \alpha_{\text{unc}}$. The controller and add effects are lifted by creating a variable $v_i$ for every distinct object that appears in $\underline{C} \cup \underline{E}^+$. The operator's arguments $\overline{v}$ are set to these variables.

Next, we must induce the preconditions and delete effects of this new operator $\omega_{\text{new}}$. To this end, we add $\omega_{\text{new}}$ to our current candidate set, and partition all data in our training set $\mathcal{D}$ into operator specific datasets $\mathcal{D}_\omega$ for each operator $\omega$ in our current candidate set. Since operator preconditions and delete effects depend on the partitioning, we first remove these from all operators that are not $\omega_{\text{new}}$ (REMOVEPRECANDDELEFFS). We perform this partitioning by running the FINDBESTCONSISTENTOP method from Algorithm 8 on this new operator set for every transition in the dataset, though we do not check the condition $s_{i+1}^{pred} \subseteq s_{i+1}$, since the operators do not yet have delete effects specified. While performing this step, we save a mapping $\delta_{\tau_i}$ from the operator's arguments to the specific objects used to ground it for every transition in the dataset (this will be used for lifting the preconditions and delete effects of each operator below). We assign each transition to the dataset associated with the operator returned by FINDBESTCONSISTENTOP. We return the operator specific datasets $(\mathcal{D}_{\omega_1} \ldots \mathcal{D}_{\omega_l})$, as well as the saved object mappings for each transition $(\delta_{\tau_1}, \ldots, \delta_{\tau_j})$.

We now induce preconditions and delete effects using $(\mathcal{D}_{\omega_1} \ldots \mathcal{D}_{\omega_l})$ and $(\delta_{\tau_1}, \ldots, \delta_{\tau_j})$. Before we do this, we delete any operator whose corresponding dataset is empty. Similar to Chitnis et al. [2], we set the *preconditions* to: $P \leftarrow \bigcap_{\tau = (s_i, \cdot, \cdot) \in \mathcal{D}_\omega} \delta_\tau(s_i)$. We also set the *atomic delete effects* to $E_\circ^- \leftarrow \bigcup_{\tau = (s_i, \cdot, s_{i+1}) \in \mathcal{D}_\omega} \delta_\tau(s_{i+1}) \setminus \delta_\tau(s_i)$. For every transition $(s_i, u_{i+1}, s_{i+1})$, let $s_{i+1}^{pred} = (s_i \setminus \underline{E_\circ^-}) \cup \underline{E}^+$. Then, we set $s_{\text{mispred}} = \bigcup_{\tau = (\cdot, \cdot, s_{i+1}) \in \mathcal{D}_\omega} s_{i+1}^{pred} \setminus s_{i+1}$. We induce a quantified delete effect for every *predicate* corresponding to atoms in $s_{\text{mispred}}$. We then set each operator's delete effects to be the union of $E_\circ^-$ and the quantified delete effects.

Now that all operators have preconditions and delete effects specified, we must ensure that the newly-added operator ($\omega_{\text{new}}$) is able to satisfy the necessary atoms for each of its transitions in $\mathcal{D}_{\omega_{\text{new}}}$. Recall that we set the operator's add effects to be the necessary atoms that changed in the first uncovered transition $\tau_{\text{unc}}$. Given the way partitioning is done (specifically the conditions in the FINDBESTCONSISTENTOP method in Algorithm 8), we know that these add effects must satisfy $\alpha_{i+1} \subseteq s_{i+1} \cup \underline{E}^+$ for all transitions $(s_i, u_{i+1}, s_{i+1}) \in \mathcal{D}_{\omega_{\text{new}}}$ with corresponding necessary atoms $\alpha_{i+1}$ for state $s_{i+1}$. However, the delete effects may cause the necessary atoms to become violated for certain transitions: i.e, $\alpha_{i+1} \not\subseteq (s_{i+1} \setminus \underline{E}^-) \cup \underline{E}^+$. For every such transition, we let $\alpha_{i+1}^{\text{miss}} = \alpha_{i+1} \setminus ((s_{i+1} \setminus \underline{E}^-) \cup \underline{E}^+)$. We then create a new operator $\omega_i^{\text{miss}}$ by copying all components of $\omega_{\text{new}}$, and adding lifted atoms from $\alpha_{i+1}^{\text{miss}}$ to both the preconditions and add effects. We modify the operator's arguments to contain new variables accordingly. This now ensures that the necessary atoms are not violated for any transition in $\mathcal{D}_{\omega_{\text{new}}}$. We add these new operators to the current candidate operator set.

After having added new operators to our candidate set in the above step, we must re-partition data and consequently re-induce preconditions and delete effects to match this new partitioning (lines 11-12 of Algorithm 6). We now have a new operator set that is guaranteed to cover the transition $\tau_{\text{unc}}$ that was initially uncovered. We check whether this new set achieves a lower value for the `coverage` term of our objective, and iterate the above steps until it does.

Finally, after the while loop terminates, we remove all operators from $\Omega'$ that have associated datasets that are *empty*. This corresponds exactly to removing operators that are not used in *any* abstract plan suffix computed by COMPUTECOVERAGE and are thus unnecessary for planning.

**Proof of termination** To see that the main loop of Algorithm 6 is guaranteed to terminate, consider that the operator set $\Omega'$ strictly grows larger at every loop iteration (no operators are deleted). Since

the predicates are fixed, there is a finite number of possible operators. Thus, at some finite iteration, $\Omega'$ will contain every possible operator. At this point, it *must* contain an operator that covers every transition and the loop must terminate.

**Anytime Removal of Operators with Null Data**   In the IMPROVECOVERAGE procedure as illustrated in Algorithm 6, we only prune out operators that do not have any data associated with them after the main while loop has terminated. However, we note here that we can remove such operators from the current operator set ($\Omega'$) at any time during the algorithm's loop.

This property arises because *the amount of data associated with a particular operator will only decrease over time*. To see this, note that (1) the number of operators in $\Omega'$ only increases over time, and (2) data is assigned to the 'best covering' operator as judged by our heuristic in Equation 3. Given a particular operator $\omega$ at some iteration $i$ of the loop, suppose there are $d$ transitions from $\mathcal{D}$ associated with it (i.e, $|\mathcal{D}_\omega| = d$). During future (i.e $> i$) loop iterations, new operators will be added to $\Omega'$. For any of the $d$ transitions in $\mathcal{D}_\omega$, these new operators can either be a worse match (in which case, the transition will remain in $\mathcal{D}_\omega$), or a better match (in which case, the transition will become associated with the new operator). Thus, for any operator $\omega$, once there is no longer any data associated with it, there will *never* be any data associated with it, and it will simply be pruned after the while loop terminates.

As a result, we can prune operators from our current set whenever there is no data associated with them. We do this in our implementation, since it improves our algorithm's wall-clock runtime.

### B.2   REDUCECOMPLEXITY

REDUCECOMPLEXITY($\Omega$, $\mathcal{D}$, $\mathcal{T}_{train}$)
1 | $\Omega' \leftarrow$ DELETEOPERATOR($\Omega$)
2 | $(\mathcal{D}_{\omega_1} \dots \mathcal{D}_{\omega_m}), (\delta_{\tau_1}, \dots, \delta_{\tau_j}) \leftarrow$ PARTITIONDATA($\Omega', \mathcal{D}$)
3 | $\Omega' \leftarrow$ INDUCEPRECANDDELEFFS($\Omega', (\mathcal{D}_{\omega_1} \dots \mathcal{D}_{\omega_m})$,
    $(\delta_{\tau_1}, \dots, \delta_{\tau_j})$)
4 | **return** $\Omega'$

**Algorithm 7:** Pseudocode for our `reduce-complexity` successor generator. The inputs are a set of operators $\Omega$, the set of all training demonstrations $\mathcal{D}$, and the corresponding set of trainign tasks $\mathcal{T}_{train}$. The output is a set of operators $\Omega'$ such that `complexity`($\Omega'$) $\leq$ `complexity`($\Omega$).

The pseudocode for our `reduce-complexity` generator is shown in Algorithm 7. As can be seen, the generator is rather simple: we simply delete an operator from the current set (DELETEOPERATOR) and return the remaining operators. Since we've changed the operator set, we must recompute the partitioning and re-induce preconditions and delete effects accordingly.

This generator clearly reduces the `complexity` term from our objective (Equation 1), since $|\Omega'| < |\Omega|$.

## C   Associating Transitions with Operators

FINDBESTCONSISTENTOP($(\Omega, s_i, s_{i+1}, \alpha, \mathcal{O})$)
1 | $\Omega_{\text{con}} \leftarrow \emptyset$
2 | **for** $\omega \in \Omega$ **do**
3 |     **for** $\underline{\omega} \in$ GETALLGROUNDINGS($\omega, \mathcal{O}$) **do**
4 |         $s_{i+1}^{\text{pred}} \leftarrow ((s_i \setminus \underline{E^-}) \cup \underline{E^+})$
5 |         **if** $\underline{P} \subseteq s_i$ AND $\alpha \subseteq s_{i+1}^{pred}$ AND $s_{i+1}^{pred} \subseteq s_{i+1}$ AND $\exists\theta : \underline{C}(\theta) = u_i$ **then**
6 |             $\Omega_{\text{con}} \leftarrow \Omega_{\text{con}} \cup \underline{\omega}$
7 |     **if** $\Omega_{con} \neq \emptyset$ **then**
        **return** FINDBESTCOVER($\Omega_{con}, (s_i, s_{i+1})$)

**Algorithm 8:** Pseudocode for the FindBestConsistentOp helper method used in Algorithm 1

A key component of our algorithm is the FINDBESTCOVER method from Algorithm 8, which in turn is used in Algorithm 1 and the PARTITIONDATA method of Algorithm 6. The purpose of this method is to associate a transition with a particular operator when multiple operators satisfy the conditions necessary to 'cover' it. Intuitively, we wish to assign a transition to the operator whose prediction best matches the *observed effects* in the transition. We can do this by simply measuring the discrepancy between the operator's add and delete effects, and the observed add and delete effects in the transition. We make two minor changes to this simple measure that are appropriate to our setting. First, we only use the *atomic* delete effects as part of our measure. We exclude the quantified delete effects because these exist in order to enable our operators to decline to predict particular changes in state. Second, we favor operators that correctly predict which atoms *will not* change. Recall that the ENSURENECATOMSSAT method in Algorithm 6 induces such operators by placing the same atoms in the add effects and preconditions.

Given some transition $(s_i, u_{i+1}, s_{i+1})$, and some ground operator $\underline{\omega}$ with atomic delete effects $\underline{E_\circ^-}$, our heuristic for data partitioning is represented by the score function shown in equation 3.

$$
\begin{aligned}
\underline{\mathcal{K}} &= \underline{E^+} \cap \underline{P} \\
\underline{\mathcal{C}} &= \underline{E^+} \setminus \underline{\mathcal{K}} \\
\text{score} = |\underline{\mathcal{C}} &\setminus (s_{i+1} \setminus s_i)| + \\
|(s_{i+1} \setminus s_i) &\setminus \underline{\mathcal{C}}| + \\
|(\underline{E_\circ^-} \setminus (s_i &\setminus s_{i+1}))| + \\
|(s_i \setminus s_{i+1}) &\setminus \underline{E_\circ^-}| - \underline{\mathcal{C}}
\end{aligned}
\tag{3}
$$

Once all eligible operators have been scored, we simply pick the lowest-scoring operator to associate with this transition. If multiple operators achieve the same score, we break ties arbitrarily.

## D    Learning Samplers

In addition to operators, we must also learn samplers to propose continuous parameters for controllers during plan refinement. We directly adapt existing approaches [2, 1] to accomplish this and learn one sampler per operator of the following form: $\sigma(x, o_1, \ldots, o_k) = s_\sigma(x[o_1] \oplus \cdots \oplus x[o_k])$, where $x[o]$ denotes the feature vector for $o$ in $x$, the $\oplus$ denotes concatenation, and $s_\sigma$ is the model to be learned. Specifically, we treat the problem as one of supervised learning on each of the datasets associated with each operator: $\mathcal{D}_\omega$. Recall that for every transition $(x_i, u_{i+1}, x_{i+1})$ in $\mathcal{D}_\omega$, we save a mapping $\delta : \overline{v} \to \mathcal{O}_\tau$ from the operator's arguments $\overline{v}$ to objects to ground the operator with. Recall also that every action is a hybrid controller with discrete parameters and continuous parameters $\theta$. To create a datapoint that can be used for supervised learning for the associated sampler, we can reuse this substitution to create an input vector $x[\delta_\tau(v_1)] \oplus \cdots \oplus x[\delta(v_k)]$, where $(v_1, \ldots, v_k) = \overline{v}$. The corresponding output for supervised learning is the continuous parameter vector $\theta$ in the action $u_{i+1}$.

Following previous work by Silver et al. [1] and Chitnis et al. [2], we learn two neural networks to parameterize each sampler. The first neural network takes in $x[o_1] \oplus \cdots \oplus x[o_k]$ and regresses to the mean and covariance matrix of a Gaussian distribution over $\theta$. We assume that the desired distribution has nonzero measure, but the covariances can be arbitrarily small in practice. To improve the representational capacity of this network, we learn a second neural network that takes in $x[o_1] \oplus \cdots \oplus x[o_k]$ *and* $\theta$, and returns true or false. This classifier is then used to rejection sample from the first network. To create negative examples, we use all transitions such that the controller used in the transition matches the current controller, but the transition is not in the operator's dataset $\mathcal{D}_\omega$.

# E  Additional Experiment Details

Here we provide detailed descriptions of each of experiment environments. See (§5) for high-level descriptions and the accompanying code for implementations.

## E.1  Screws Environment Details

- **Types:**
  - The `screw` type has features `x`, `y`, `held`.
  - The `receptacle` type has features `x`, `y`.
  - The `gripper` type has features `x`, `y`.

- **Predicates:** `Pickable(?x0:gripper, ?x1:receptacle)`, `AboveReceptacle(?x0:gripper, ?x1:receptacle)`, `HoldingScrew(?x0:gripper, ?x1:screw)`, `ScrewInReceptacle(?x0:screw, ?x1:receptacle)`.

- **Actions:**
  - `MoveToScrew(?x0: gripper, ?x1: screw)`: moves the gripper to be `Near` the screw ?x1.
  - `MoveToReceptacle(?x0: gripper, ?x1: receptacle)`: moves the gripper to be AboveReceptacle(?x0:gripper, ?x1:receptacle)
  - `MagnetizeGripper(?x0: gripper)`: Magnetizes the gripper at the current location, which causes all screws that the gripper is `Near` to be held by the gripper.
  - `DemagnetizeGripper(?x0: gripper)`: Demagnetizes the gripper at the current location, which causes all screws that are being held by the gripper to fall.

- **Goal:** The agent must make `ScrewInReceptacle(?x0:screw, ?x1:receptacle)` true for a particular screw that varies per task.

## E.2  Cluttered 1D Environment Details

- **Types:**
  - The `robot` type has features `x`.
  - The `dot` type has features `x`, `grasped`.

- **Predicates:** `NextTo(?x0:robot, ?x1:dot)`, `NextToNothing(?x0:robot)`, `Grasped(?x0:robot, ?x1:dot)`.

- **Actions:**
  - `MoveGrasp(?x0: robot, ?x1: dot, [move_or_grasp, x])`: A single controller that performs both moving and grasping. If move_or_grasp $< 0.5$, then the controller moves the robot to a continuous position `y`. Else, the controller grasps the dot ?x1 if it is within range.

- **Goal:** The agent must make `Grasped(?x0:robot, ?x1:dot)` true for a particular set of dots that varies per task.

## E.3  Satellites Environment Details

- **Types:**
  - The `satellite` type has features `x`, `y`, `theta`, `instrument`, `calibration_obj_id`, `is_calibrated`, `read_obj_id`, `shoots_chem_x`, `shoots_chem_y`.

– The `object` type has features `id`, `x`, `y`, `has_chem_x`, `has_chem_y`.

- **Predicates:** `Sees(?x0:satellite, ?x1:object)`, `CalibrationTarget(?x0:satellite, ?x1:object)`, `IsCalibrated(?x0:satellite)`, `HasCamera(?x0:satellite)`, `HasInfrared(?x0:satellite)`, `HasGeiger(?x0:satellite)`, `ShootsChemX(?x0:satellite)`, `ShootsChemY(?x0:satellite)`, `HasChemX(?x0:satellite)`, `HasChemY(?x0:satellite)`, `CameraReadingTaken(?x0:satellite, ?x1:object)`, `InfraredReadingTaken(?x0:satellite, ?x1:object)`, `GeigerReadingTaken(?x0:satellite, ?x1:object)`.

- **Actions:**

  - `MoveTo(?x0:satellite, ?x1:object, [x, y])`: Moves the satellite `?x0` to be at `x, y`.
  - `Calibrate(?x0:satellite, ?x1:object)`: Tries to calibrate the satellite `?x0` against object `?x1`. This will only succeed (i.e, make `IsCalibrated(?x0:satellite)` true) if `?x1` is the calibration target of `?x0`.
  - `ShootChemX(?x0:satellite, ?x1: object)`: Tries to shoot a pellet of chemical X from satellite `?x0`. This will only succeed if `?x0` both has chemical X and is capable of shooting it.
  - `ShootChemY(?x0:satellite, ?x1:object)`: Tries to shoot a pellet of chemical Y from satellite `?x0`. This will only succeed if `?x0` both has chemical Y and is capable of shooting it.
  - `UseInstrument(?x0:satellite, ?x1:object)`: Tries to use the instrument possessed by `?x0` on object `?x1` (note that we assume `?x0` only possesses a single instrument).

- **Goal:** The agent must take particular readings (i.e some combination of `CameraReadingTaken(?x0:satellite, ?x1:object)`, `InfraredReadingTaken(?x0:satellite, ?x1:object)`, `GeigerReadingTaken(?x0:satellite, ?x1:object)`) from a specific set of objects that varies per task.

### E.4 Painting Environment Details

- **Types:**

  - The `object` type has features `x`, `y`, `z`, `dirtiness`, `wetness`, `color`, `grasp`, `held`.
  - The `box` type has features `x`, `y`, `color`.
  - The `lid` type has features `open`.
  - The `shelf` type has features `x`, `y`, `color`.
  - The `robot` type has features `x`, `y`, `fingers`.

- **Predicates:** `InBox(?x0:obj)`, `InShelf(?x0:obj)`, `IsBoxColor(?x0:obj, ?x1:box)`, `IsShelfColor(?x0:obj, ?x1:shelf)`, `GripperOpen(?x0:robot)`, `OnTable(?x0:obj)`, `NotOnTable(?x0:obj)`, `HoldingTop(?x0:obj)`, `HoldingSide(?x0:obj)`, `Holding(?x0:obj)`, `IsWet(?x0:obj)`, `IsDry(?x0:obj)`, `IsDirty(?x0:obj)`, `IsClean(?x0:obj)`.

- **Actions:**

- – `Pick(?x0:robot, ?x1:obj, [grasp])`: picks up a particular object, if grasp $> 0.5$ it performs a top grasp otherwise a side grasp.
    - – `Wash(?x0:robot)`: washes the object in hand, which is needed to clean the object.
    - – `Dry(?x0:robot)`: drys the object in hand, which is needed after you wash the object.
    - – `Paint(?x0:robot, [color])`: paints the object in hand a particular color specified by the continuous parameter.
    - – `Place(?x0:robot, [x, y, z])`: places the object in hand at a particular x, y, z location specified by the continuous parameters.
    - – `OpenLid(?x0:robot, ?x1:lid)`: opens a specific lid, which is need to place objects inside the box.

- **Goal:** A robot in 3D must pick, wash, dry, paint, and then place various objects in order to get `InBox(?x0:obj)` and `IsBoxColor(?x0:obj, ?x1:box)`, or `InShelf(?x0:obj)` and `IsShelfColor(?x0:obj, ?x1:shelf)` true for particular goal objects.

## E.5 Cluttered Painting Environment Details

- **Types:**
    - – The `object` type has features x, y, z, `dirtiness`, `wetness`, `color`, `grasp`, `held`.
    - – The `box` type has features x, y, `color`.
    - – The `lid` type has features `open`.
    - – The `shelf` type has features x, y, `color`.
    - – The `robot` type has features x, y, `fingers`.

- **Predicates:** `InBox(?x0:obj)`, `InShelf(?x0:obj)`, `IsBoxColor(?x0:obj, ?x1:box)`, `IsShelfColor(?x0:obj, ?x1:shelf)`, `GripperOpen(?x0:robot)`, `OnTable(?x0:obj)`, `NotOnTable(?x0:obj)`, `HoldingTop(?x0:obj)`, `HoldingSide(?x0:obj)`, `Holding(?x0:obj)`, `IsWet(?x0:obj)`, `IsDry(?x0:obj)`, `IsDirty(?x0:obj)`, `IsClean(?x0:obj)`, along with RepeatedNextTo Predicates: `NextTo(?x0:robot, ?x1:obj)`, `NextToBox(?x0:robot, ?x1:box)`, `NextToShelf(?x0:robot, ?x1:shelf)`, `NextToTable(?x0:robot, ?x1:table)`.

- **Actions:**
    - – `Pick(?x0:robot, ?x1:obj, [grasp])`: picks up a particular object, if grasp $> 0.5$ it performs a top grasp otherwise a side grasp.
    - – `Wash(?x0:robot)`: washes the object in hand, which is needed to clean the object.
    - – `Dry(?x0:robot)`: drys the object in hand, which is needed after you wash the object.
    - – `Paint(?x0:robot, [color])`: paints the object in hand a particular color specified by the continuous parameter.
    - – `Place(?x0:robot, [x, y, z])`: places the object in hand at a particular x, y, z location specified by the continuous parameters.
    - – `OpenLid(?x0:robot, ?x1:lid)`: opens a specific lid, which is need to place objects inside the box.
    - – `MoveToObj(?x0:robot, ?x1:obj, [x])`: moves to a particular object with certain displacement x.

- MoveToBox(?x0:robot, ?x1:box, [x]): moves to a particular box with certain displacement x.
- MoveToShelf(?x0:robot, ?x1:shelf, [x]): moves to a particular shelf with certain displacement x.

- **Goal:** A robot in 3D must pick, wash, dry, paint, and then place various objects in order to get InBox(?x0:obj) and IsBoxColor(?x0:obj, ?x1:box), or InShelf(?x0:obj) and IsShelfColor(?x0:obj, ?x1:shelf) true for particular goal objects. In contrast to the previous painting environment, we also need to navigate to the right objects (i.e. all objects are not always reachable from any states). This version of the environment requires operators with quantified delete effects.

### E.6 BEHAVIOR Environment Details

- **Types:**
  - Many object types that range from relevant types like hardbacks and notebooks to many irrelevant types like toys and jars. All object types have features from location and orientation to graspable and open. For a complete list of object types and features see [5].

- **Predicates:** Inside(?x0:obj, ?x1:obj), OnTop(?x0:obj, ?x1:obj), Reachable-Nothing(), HandEmpty(), Holding(?x0:obj), Reachable(), Openable(?x0:obj), Not-Openable(?x0:obj), Open(?x0:obj), Closed(?x0:obj).

- **Actions:**
  - NavigateTo(?x0:obj): navigates to make a particular object reachable.
  - Grasp(?x0:obj, [x, y, z]): picks up a particular object with the hand starting at a particular relative x, y, z location specified by the continuous parameters.
  - PlaceOnTop(?x0:obj): places the object in hand ontop of another object as long as the agent is holding an object and is in range of the object to be placed onto.
  - PlaceInside(?x0:obj): places the object in hand inside another object as long as the agent is holding an object and is in range of the object to be placed into.
  - Open(?x0:obj)): opens a specific object (windows, doors, boxes, etc.) if it is 'openable'.
  - Close(?x0:obj)): closes a specific object (windows, doors, boxes, etc.) if it is currently in an 'open' state.

- **Goal:** In *Opening Presents*, the robot must Open(?x0:package) a number of boxes of type package around the room. In *Locking Windows*, the robot must navigate around the house to Close(?x0:window) a number of windows. In *Collecting Cans*, the robot must pick up a number of empty soda cans of type pop strewn amongst the house and throw them into a trash can of type bucket. This will satisfy the goal of getting Inside(?x0:pop, ?x1:bucket) for every soda can around the house. In *Sorting Books*, the robot must find books of type hardback and notebook in a living room and place them each onto a cluttered shelf (i.e. satisfy the goal of OnTop(?x0:hardback, ?x1:shelf) and OnTop(?x0:notebook, ?x1:shelf) for a number of books).

## F  Additional Approach Details

Here we provide detailed descriptions of each approach evaluated in experiments. For the approaches that learn operators, we use $A^*$ search with the lmcut heuristic [44] as the high-level planner for bilevel planning in non-BEHAVIOR environments, and use Fast Downward [11] in a

configuration with minor differences from `lama-first` as the high-level planner in BEHAVIOR environments, since $A^*$ search was unable to find abstract plans given the large state and action spaces of these tasks. All approaches also iteratively resample until the simulator $f$ verifies that the transition has been achieved, except for GNN Model-Free, which is completely model-free. See (§5) for high-level descriptions and the accompanying code for implementations.

## F.1 Ours

- **Operator Learning:** We learn operators via the hill-climbing search described in Section 4.3. For our objective (Equation 1), we set the $\lambda$ term to be $1/|\mathcal{D}|$, where $|\mathcal{D}|$ represents the number of transitions in the training demonstrations.

- **Sampler Learning:** As described in Section D, each sampler consists of two neural networks: a generator and a discriminator. The generator outputs the mean and diagonal covariance of a Gaussian, using an exponential linear unit (ELU) to assure PSD covariance. The generator is a fully-connected neural network with two hidden layers of size 32, trained with Adam for 50,000 epochs with a learning rate of $1e-3$ using Gaussian negative log likelihood loss. The discriminator is a binary classifier of samples output by the generator. Negative examples for the discriminator are collected from other skill datasets. The classifier is a fully-connected neural network with two hidden layers of size 32, trained with Adam for 10,000 epochs with a learning rate of $1e-3$ using binary cross entropy loss. During planning, the generator is rejection sampled using the discriminator for up to 100 tries, after which the last sample is returned.

- **Planning:** The number of abstract plans for high-level planning was set to $N_{\text{abstract}} = 8$ for our non-BEHAVIOR domains, and $N_{\text{abstract}} = 1$ for our BEHAVIOR domains. The samples per step for refinement was set to $N_{\text{samples}} = 10$ for all environments.

## F.2 Cluster and Intersect:

This is the operator learning approach used by Silver et al. [1].

- **Operator Learning:** This approach learns STRIPS operators by attempting to induce a different operator for every set of unique lifted effects (See Silver et al. [1] for more information).

- **Sampler Learning and Planning:** Same as Ours (See (§F.1) for more details).

## F.3 LOFT:

This is the operator learning approach used by Silver et al. [3]. We include a version ('LOFT+Replay') that is allowed to mine additional negative data from the environment to match the implementation of the original authors. We also include a version ('LOFT') that is restricted to learning purely from the demonstration data.

- **Operator Learning:** This approach learns operators similar to the Cluster and Intersect baseline, except that it uses search to see if it can modify the operators after performing Cluster and Intersect (See Silver et al. [3] for more information).

- **Sampler Learning and Planning:** Same as Ours (See (§F.1) for more details).

## F.4 CI + QE:

A baseline variant of Cluster and Intersect that is capable of learning operators that have quantified delete effects in addition to atomic delete effects.

- **Operator Learning:** This approach first runs Cluster and Intersect, then attempts to induce quantified delete effects by performing a hill-climbing search over possible choices of quantified delete effects using prediction error as the metric to be optimized.

- **Sampler Learning and Planning:** Same as Ours (See (§F.1) for more details).

## F.5 GNN Shooting:

This approach trains a graph neural network (GNN) [45] policy. This GNN takes in the current state $x$, abstract state $s = \text{ABSTRACT}(x, \Psi_G)$, and goal $g$. It outputs an action via a one-hot vector over $\mathcal{C}$ corresponding to which controller to execute, one-hot vectors over all objects at each discrete argument position, and a vector of continuous arguments. We train the GNN using behavior cloning on the dataset $\mathcal{D}$. At evaluation time, we sample trajectories by treating the GNN's output continuous arguments as the mean of a Gaussian with fixed variance. We use the known transition model $f$ to check if the goal is achieved, and repeat until the planning timeout is reached.

- **Planning:** Repeat until the goal is reached: query the model on the current state, abstract state, and goal to get a ground skill. Invoke the ground skill's sampler up to 100 times to find a subgoal that leads to the abstract successor state predicted by the skill's operator. If successful, simulate the state forward; otherwise, terminate with failure.

- **Learning:** This approach essentially learns a TAMP planner in the form of a GNN. Following the baselines presented in prior work [2], the GNN is a standard encode-process-decode architecture with 3 message passing steps. Node and edge modules are fully-connected neural networks with two hidden layers of size 16. We follow the method of Chitnis et al. [2] for encoding object-centric states, abstract states, and goals into graph inputs. To get graph outputs, we use node features to identify the object arguments for the skill and a global node with a one-hot vector to identify the skill identity. The models are trained with Adam for 1000 epochs with a learning rate of $1e-3$ and batch size 128 using MSE loss.

## F.6 GNN Model-Free:

A baseline that uses the same trained GNN as above, but at evaluation time, directly executes the policy instead of checking execution using $f$. This has the advantage of being more efficient to evaluate than GNN Shooting, but is less effective.

# G    Additional Experimental Results and Analyses

| Environment | Ours | LOFT | LOFT+replay | CI | CI + QE | GNN |
|---|---|---|---|---|---|---|
| Painting | **69.35 (3.58)** | 92.26 (11.41) | 135.73 (6.45) | **70.95 (5.07)** | **67.08 (5.86)** | 2220.19 (181.29) |
| Satellites | **19.38 (7.83)** | 52.73 (18.35) | 438.44 (51.62) | 23.29 (5.38) | **15.96 (4.70)** | 1625.69 (218.88) |
| Clutter 1D | **17.98 (1.06)** | 68.04 (17.68) | 366.89 (146.09) | 62.68 (14.89) | 28.58 (3.68) | 1164.92 (84.74) |
| Screws | 1.31 (0.04) | 143.60 (49.10) | 5712.80 (736.84) | **0.32 (0.02)** | 708.98 (1023.02) | 1369.59 (68.44) |
| Cluttered Satellites | **16.12 (0.55)** | 353.67 (52.78) | 902.99 (148.22) | 107.04 (11.94) | 87.24 (10.49) | 3043.62 (285.27) |
| Cluttered Painting | **131.68 (5.05)** | 1699.52 (216.71) | 7364.03 (532.67) | 470.32 (40.38) | 2788.74 (1330.38) | 4615.70 (334.11) |
| Opening Presents | **28.91 (11.26)** | 106.57 (27.72) | - | 100.62 (23.66) | 92.63 (17.01) | 185.53 (6.63) |
| Locking Windows | **16.77 (1.55)** | 62.55 (10.12) | - | 61.71 (8.95) | 45.51 (5.74) | 319.09 (7.61) |
| Collecting Cans | 3728.73 (9544.75) | 1520.93 (354.20) | - | **576.89 (100.57)** | 781.38 (350.46) | 2121.86 (120.51) |
| Sorting Books | 4981.79 (14460.37) | 6423.03 (602.44) | - | **1528.18 (111.18)** | - | 5359.99 (170.46) |

Table 2: Learning times in seconds on training data for all domains. Note that BEHAVIOR domains (bottom 4) use training set sizes of 10 tasks, while all other domains use training and testing set sizes of 50 tasks. The standard deviation is shown in parentheses.

We have already established that our approach learns operators that lead to more effective *bilevel planning* than baselines. In this section, we are interested in comparing our approach with baselines on three additional metrics: (1) the efficiency of high-level planning using learned operators, (2) the efficiency of the learning algorithm itself, and (3) the simplicity of operator sets we learn. We also run ablations of our method to investigate the importance of optimizing the complexity term, as well as downward-refinability to our method.

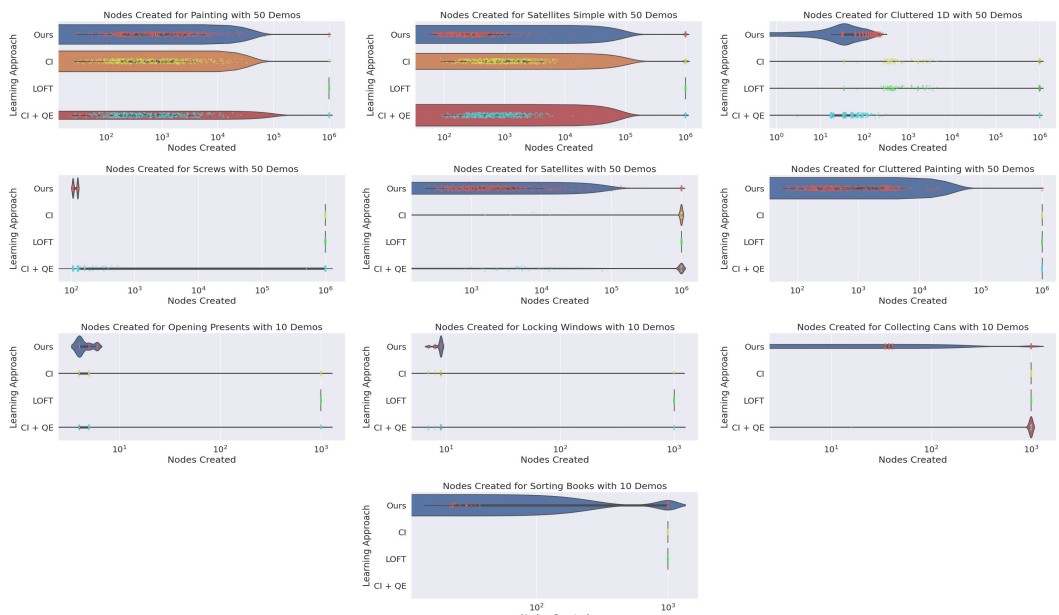

Figure 4: **Nodes Created by Operator Learning Approaches**. We show scatter plots of the nodes created (x-axis) for each operator learning approach (y-axis). We also include a violin graph to visualize the density of points throughout the graph. If bilevel planning failed, we set the nodes created to $10^6$ for non-BEHAVIOR domains and $10^3$ for BEHAVIOR domains. Our approach achieves a low number of nodes created across when compared to baselines in most domains.

| Environment | Ours | LOFT | LOFT+replay | CI | CI + QE |
|---|---|---|---|---|---|
| Painting | **10.00 (0.00)** | 13.60 (0.80) | 19.20 (0.39) | 11.00 (0.00) | 10.20 (0.40) |
| Satellites | **7.40 (0.79)** | 10.90 (1.44) | 33.80 (3.70) | 10.40 (1.20) | 9.30 (0.9) |
| Clutter 1D | **2.00 (0.00)** | 7.10 (1.64) | 16.10 (2.11) | 7.10 (1.64) | 3.00 (0.44) |
| Screws | **4.0 (0.00)** | 14.80 (1.98) | 91.14 (5.11) | 14.80 (1.98) | 4.80 (0.97) |
| Cluttered Satellites | **7.00 (0.00)** | 19.80 (2.60) | 59.60 (4.45) | 16.30 (1.10) | 13.9 (0.83) |
| Cluttered Painting | **13.00 (0.00)** | 28.00 (0.00) | 157.8 (6.49) | 25.20 (2.31) | 20.70 (1.61) |
| Opening Presents | **2.30 (0.90)** | 10.80 (2.99) | - | 10.80 (2.99) | 9.80 (1.83) |
| Locking Windows | **2.00 (0.00)** | 6.10 (0.70) | - | 6.10 (0.70) | 4.70 (0.64) |
| Collecting Cans | **6.10 (5.37)** | 57.40 (9.43) | - | 52.90 (8.41) | 13.40 (2.33) |
| Sorting Books | **14.70 (7.57)** | 76.70 (5.62) | - | 75.80 (5.79) | - |

Table 3: Average number of operators learned for all domains. Note that BEHAVIOR domains (bottom 4) use training set sizes of 10 tasks, while all other domains use training and testing set sizes of 50 tasks. The standard deviation is shown in parentheses.

Figure 4 shows the nodes created during high-level planning for each of our various environments and operator learning methods. We can see that operators learned by our approach generally lead to comparable or fewer node creations during planning when compared to baselines. In many of the environments where baseline methods are able to achieve a number of points with fewer node creations — *Cluttered 1D*, *Opening Presents*, and *Locking Windows* — our method has a significantly higher success rate.

Table 2 shows the learning times for all methods in all domains[3]. Our approach achieves the lowest learning time in 7/10 domains. Upon inspection of our method's performance on the 'Locking Windows' and 'Collecting Cans' domains, we discovered that the high average learning times are because of a few outlier seeds encountering local minima learning, yielding large and complex operator sets (this is the reason for the extremely high standard deviation).

---

[3]Note that there is no entry for 'CI + QE' for sorting books because learning exceeded the memory limit of our hardware (192 GB)

| Environment | Ours | No complexity | Down Eval |
|---|---|---|---|
| Painting | 98.80 (1.33) | 98.80 (1.33) | 26.60 (6.52) |
| Satellites | 93.40 (11.14) | 81.20 (19.40) | 84.20 (16.88) |
| Cluttered 1D | 100.00 (0.00) | 100.0 (0.00) | 100.00 (0.00) |
| Screws | 100.00 (0.00) | 100.00 (0.00) | 100.00 (0.00) |
| Cluttered Satellites | 95.20 (2.40) | 94.80 (2.72) | 94.00 (3.22) |
| Cluttered Painting | 99.20 (1.33) | 99.00 (1.84) | 23.80 (7.56) |
| Opening Presents | 100.00 (0.00) | 100.00 (0.00) | 100.00 (0.00) |
| Locking Windows | 100.00 (0.00) | 100.00 (0.00) | 100.00 (0.00) |
| Collecting Cans | 77.00 (37.16) | 75.00 (38.30) | 75.00 (38.80) |
| Sorting Books | 69.00 (36.73) | 52.00 (34.00) | 67.00 (37.2) |

Table 4: Percentage success rate for our original method, as well as ablations where we set the $\lambda$ parameter from Equation 1 to 0 on test tasks (No complexity), and enforce downward refinability at evaluation time (Down Eval) for all domains. Note that BEHAVIOR domains use training and testing set sizes of 10 tasks, while all other domains use training and testing set sizes of 50 tasks. The percentage standard deviation is shown in brackets.

Table 3 shows the number of operators learned for all operator learning methods in all domains. Our approach learns the lowest number of operator sets across all environments and massively out performs other approaches on this metric in *Collecting Cans*, and *Sorting Books*. These results further highlight our ability to learn operator sets that efficient for high-level planning, but also simpler and therefore, more likely to generalize to new environments.

Table 4 shows the success rate of our method when the $\lambda$ parameter from Equation 1 is set to 0 (the 'No complexity' column), thereby effectively removing any impact on the optimization from the `complexity` term in the objective. In most environments (Painting, Cluttered 1D, Screws, Cluttered Satellites, Cluttered Painting, Opening Presents, Locking Windows, and Collecting Cans), this change has minimal impact on the method's success rate. However, in two environments (Satellites and Sorting Books), the change causes a significant reduction in success rate. Upon inspection, we found that our approach learned a number of very complex operators in these domains. When $\lambda$ was set to be greater than 0, our approach would delete a number of these operators, but in this ablation, it was unable to and thus planning performance suffered. We can conclude from these experiments that optimizing the complexity term is a key component of our approach on particular domains. In fact, we believe that more aggressively optimizing the complexity term, perhaps by increasing the sophistication of our REDUCECOMPLEXITY step, could enable us to improve performance significantly even on the two complex BEHAVIOR tasks, since we found that our approach learned overly-complex operators for a few random seeds.

Table 4 also shows the success rate of our method when bilevel planning is only allowed to produce 1 abstract plan during refinement. This effectively enforces that the learned operators must yield downward-refinable plans. This causes a significant reduction in success rate in many environments, showing that the ability to evaluate multiple abstract plans is important to planning with operators learned by our method. Indeed, all the environments that exhibit significant reductions in success rate do not - to our knowledge - admit a downward-refinable high-level theory over the provided skills and predicates.

## H   Learned Operator Examples

Finally, we provide operator examples to demonstrate our approaches ability to overcome overfitting to specific situations. Figure 5 shows a comparison of the operators learned with *Open* in *Opening Packages* environment and *NavigateTo* in *Collecting Cans* environment across our approach and 'CI + QE' (the most competitive baseline in these environments). As shown, by optimizing prediction error 'CI + QE' learns a number of operators to describe the same amount of transitions that is covered by the single operator our approach learns. Upon inspection, 'CI + QE' learns overly specific operators when trying to cluster effects that try to predict the entire state to the point where 'Quan-

```
Open-package0:                              Open-package0:
    Arguments: [?x0:package]                    Arguments: [?x0:package]
    Preconditions: [                            Preconditions: [
        closed-package(?x0:package),                closed-package(?x0:package),
        handempty(),                                handempty(),
        openable-package(?x0:package),              openable-package(?x0:package),
        reachable-package(?x0:package)]             reachable-package(?x0:package)]
    Add Effects: [open-package(?x0:package)]    Add Effects: [open-package(?x0:package)]
    Delete Effects: [closed-package(?x0:package)]  Delete Effects: [closed-package(?x0:package)]
    Quantified Delete Effects:                  Quantified Delete Effects: []
        [ontop-package-room_floor]              Controller: Open-package(?x0:package),
    Controller: Open-package(?x0:package),

                                            Open-package1:
                                                Arguments: [?x0:room_floor, ?x1:package]
                                                Preconditions: [
                                                    closed-package(?x1:package),
                                                    handempty(),
                                                    not-openable-room_floor(?x0:room_floor),
                                                    ontop-package-room_floor(?x1:package, ?x0:room_floor),
                                                    openable-package(?x1:package),
                                                    reachable-package(?x1:package),
                                                    reachable-room_floor(?x0:room_floor)]
                                                Add Effects: [open-package(?x1:package)]
                                                Delete Effects: [closed-package(?x1:package),
                                                    ontop-package-room_floor(?x1:package, ?x0:room_floor)]
                                                Quantified Delete Effects: []
                                                Controller: Open-package(?x1:package)}
─────────────────────────────────────────────────────────────────────────────────────
NavigateTo-pop0:                            NavigateTo-pop0:
    Arguments: [?x0:pop]                         Arguments: [?x0:bed, ?x1:pop]
    Preconditions: [                            Preconditions: [
        handempty(),                                handempty(),
        not-openable-pop(?x0:pop)]                  not-openable-bed(?x0:bed),
    Add Effects: [reachable-pop(?x0:pop)]           not-openable-pop(?x1:pop),
    Delete Effects: []                              ontop-pop-bed(?x1:pop, ?x0:bed),
    Quantified Delete Effects: [                    reachable-bed(?x0:bed)]
        ontop-pop-pop,                          Add Effects: [reachable-pop(?x1:pop)]
        reachable-bed,                          Delete Effects: [reachable-bed(?x0:bed)]
        reachable-bucket,                       Quantified Delete Effects: []
        reachable-pop]                          Controller: NavigateTo-pop(?x1:pop),
    Controller: NavigateTo-pop(?x0:pop)}
                                            NavigateTo-pop1:
                                                Arguments: [?x0:pop]
                                                Preconditions: [
                                                    handempty(),
                                                    not-openable-pop(?x0:pop),
                                                    reachable-pop(?x0:pop)]
                                                Add Effects: []
                                                Delete Effects: []
                                                Quantified Delete Effects: []
                                                Controller: NavigateTo-pop(?x0:pop),

                                            NavigateTo-pop2:
                                                Arguments: [?x0:pop]
                                                Preconditions: [
                                                    handempty(),
                                                    not-openable-pop(?x0:pop)]
                                                Add Effects: [reachable-pop(?x0:pop)]
                                                Delete Effects: []
                                                Quantified Delete Effects: []
                                                Controller: NavigateTo-pop(?x0:pop),
```

Figure 5: **Operator Comparison**. Operators learned after our approach (left) and 'CI+QE' (right), for *Open* in *Opening Packages* environment (top) and *NavigateTo* in *Collecting Cans* Cans environment. Our approach learns fewer operators that are generally simpler, and thus more conducive to effective bilevel planning and generalization.

tified Delete Effects' are not fully utilized. For the full set of operators learned by our algorithm on the 'Sorting Books' task, see Figure 6.

```
Grasp-notebook0:
    Arguments: [?x0:notebook]
    Preconditions: [handempty(), not-openable-notebook(?x0:notebook), reachable-notebook(?x0:notebook)]
    Add Effects: [holding-notebook(?x0:notebook)]
    Delete Effects: [handempty(), reachable-notebook(?x0:notebook)]
    Quantified Delete Effects: [ontop-notebook-coffee_table, ontop-notebook-room_floor]
    Controller: Grasp-notebook(?x0:notebook)

NavigateTo-notebook0:
    Arguments: [?x0:notebook]
    Preconditions: [handempty(), not-openable-notebook(?x0:notebook)]
    Add Effects: [reachable-notebook(?x0:notebook)]
    Delete Effects: []
    Quantified Delete Effects: [reachable-board_game, reachable-coffee_table, reachable-hardback, reachable-notebook, reachable-shelf, reachable-video_game]
    Controller: NavigateTo-notebook(?x0:notebook)

PlaceOnTop-shelf0:
    Arguments: [?x0:shelf, ?x1:hardback]
    Preconditions: [holding-hardback(?x1:hardback), not-openable-hardback(?x1:hardback), not-openable-shelf(?x0:shelf), reachable-shelf(?x0:shelf)]
    Add Effects: [handempty(), ontop-hardback-shelf(?x1:hardback, ?x0:shelf)]
    Delete Effects: [holding-hardback(?x1:hardback)]
    Quantified Delete Effects: []
    Controller: PlaceOnTop-shelf(?x0:shelf)

PlaceOnTop-shelf1:
    Arguments: [?x0:shelf, ?x1:notebook]
    Preconditions: [holding-notebook(?x1:notebook), not-openable-notebook(?x1:notebook), not-openable-shelf(?x0:shelf), reachable-shelf(?x0:shelf)]
    Add Effects: [handempty(), ontop-notebook-shelf(?x1:notebook, ?x0:shelf)]
    Delete Effects: [holding-notebook(?x1:notebook)]
    Quantified Delete Effects: []
    Controller: PlaceOnTop-shelf(?x0:shelf)

Grasp-hardback0:
    Arguments: [?x0:hardback]
    Preconditions: [handempty(), not-openable-hardback(?x0:hardback), reachable-hardback(?x0:hardback)]
    Add Effects: [holding-hardback(?x0:hardback)]
    Delete Effects: [handempty(), reachable-hardback(?x0:hardback)]
    Quantified Delete Effects: [ontop-hardback-coffee_table, ontop-hardback-room_floor]
    Controller: Grasp-hardback(?x0:hardback)

NavigateTo-shelf0:
    Arguments: [?x0:shelf]
    Preconditions: [not-openable-shelf(?x0:shelf)]
    Add Effects: [reachable-shelf(?x0:shelf)]
    Delete Effects: []
    Quantified Delete Effects: [ontop-hardback-coffee_table, reachable-board_game, reachable-coffee_table, reachable-hardback, reachable-notebook,
reachable-video_game]
    Controller: NavigateTo-shelf(?x0:shelf)

NavigateTo-hardback0:
    Arguments: [?x0:hardback]
    Preconditions: [handempty(), not-openable-hardback(?x0:hardback)]
    Add Effects: [reachable-hardback(?x0:hardback)]
    Delete Effects: []
    Quantified Delete Effects: [reachable-board_game, reachable-coffee_table, reachable-hardback, reachable-notebook, reachable-shelf, reachable-video_game]
    Controller: NavigateTo-hardback(?x0:hardback)
```

Figure 6: Sorting Books learned operators.

