# OpenReview forum: "Learning Efficient Abstract Planning Models that Choose What to Predict"
_robot-learning.org/CoRL/2023/Conference — CoRL 2023 Poster_

### Official Review · Reviewer_uCVm · 2023-07-13

**Confidence:** 5
**Originality:** Very Good
**Technical Quality:** Excellent
**Clarity Of Presentation:** Very Good
**Impact:** 3

**Recommendation:**

Weak Accept: I recommend accepting the paper, but will not argue for my recommendation if the majority of other reviewers have a different opinion.

**Review:**

- The paper is well-written, with clear motivation, a method that well-addresses the motivation, extensive experiments, and experimental results that show the proposed method has significant quantitative and qualitative advantages.
- Learning symbolic operators from experience in a self-supervised way is a very challenging problem. It is also crucial as it enables efficient AI planners in robots operating in a continuous sensorimotor world. Learning from demonstrations (rather than unorganized action executions) makes it easier to learn, but still, it is a challenging task that is rarely studied by researchers.
- The authors argue that the core contribution of the method is to "learn operators based on planning performance rather than prediction accuracy." They cite [1] and even use [1] as one of the baselines. However, [1] also discusses that operators should be learned based on their planning performance rather than prediction performance. [1] also uses a planning-based metric in learning operators. The authors did not mention this. Although the performance of the current method is shown to be significantly better than [1], this aspect makes the theoretical contribution of the current paper look weaker.
- The authors mention that F encodes a non-deterministic abstract transition model. However, the learned operators are deterministic.
- The authors argue that their method "strongly generalizes over initial states, actions, and goals" and "generalizes to held-out" tasks. I could not find a justification for this claim in the main manuscript or the Appendix.

**Quality Of The Limitations Section:**

Limitations are addressed clearly

**Questions For Rebuttal:**

- Please clearly address and discuss the planning-metric based approach followed in [1] and rephrase the main contributions of this paper. Please also include a discussion in the experimental evaluation section as well.
- Please provide an ablation where the method only relies on the backchaining procedure (the current method without hill-climbing).
- Please provide justification for the generalization argument.
- Please provide an ablation where the method only relies on the backchaining procedure (the current method without hill-climbing).
- Please provide justification for the generalization arguments.

**Robotics Focus:**

Highly relevant to robotics but no hardware experiments

**Summary Of Paper:**

This paper proposed a method that learns symbolic operators (a list of symbolic pre-conditions and effects for high-level actions). The set of possible predicates that can appear in the operators are assumed to be known. The operators are extracted from given demonstrations that have explicit goals.

The main contributions are:
- a backchaining procedure that identifies relevant predicates in the demonstrations (different steps of the demonstrations include a large number of predicates that are irrelevant to the task and that need to be filtered out),
- a hill-climbing search that identifies the best set of operators to achieve planning best, and
- a rigorous verification of the method in many complex simulated environments compared with strong baselines.

The main argument of the paper is learning the operators based on their planning performance rather than their prediction accuracy is necessary and bring high gain in planning.

**Summary Of Recommendation:**


In short, the paper studies an important and challenging problem, proposes a novel technical solution, rigorously verifies the method in a wide range of tasks and environments, and shows that the proposed method significantly outperforms a number of baselines, including [1], which is SOTA and the closest competitor. The theoretical contribution can be highlighted better.

Update: The novelty of planning metric is more clear now. This work is not incremental but uses an idea that was already explored before. If the planning-driven learning idea were first used in this paper, I would have recommended strong acceptance. Now, I will keep my weak accept recommendation.

---

### Official Review · Reviewer_sXyD · 2023-07-18

**Confidence:** 4
**Originality:** Very Good
**Technical Quality:** Excellent
**Clarity Of Presentation:** Good
**Impact:** 4

**Recommendation:**

Strong Accept: I recommend accepting the paper and will argue for my recommendation even if other reviewers hold a different opinion.

**Review:**

This paper represents a solid technical contribution to bi-level planning. Its central insight (that you can get away with operator descriptions that only track necessary changes rather than the full set of abstract state changes) is novel, and has a solid grounding both in its mathematical description and experimental results.

The experiments themselves are thorough, with a wide range of benchmarks and sensible recent baselines. One thing that bothered me a little: It seems that the "generalization" of the learned operators depends pretty heavily on *which* demonstrations it is trained on, and what the goals of those demonstrations were. I would have liked to see a little discussion or experiments on this (Though to be fair, they do mention a dependency on the *number* of demonstrations...just not which).

Additionally, for two of the experiments (Collecting Cans and Sorting books), their algorithm gets around a 30% failure rate. The analysis concludes that this is just "some random seeds for these tasks get stuck in local minima", but I am wondering if there is more to say about this? What is it about these tasks that make them more susceptible to getting stuck in local minima? Is this something that their algorithm might be able to overcome in some way?

I also noted that, very briefly in the related literature and a sentence in the experimental results, the authors mention the concept of "downward refinability" as a critical assumption used by many bi-level planners. They mention that their algorithm does not depend on this assumption and thus has a big advantage. I was surprised therefore to not see any mention of this concept in the main methods section of the paper, or how it arises in their algorithm.

My main small issue with this work is clarity: Particularly in sections 3 & 4, I found it a struggle to keep hold of an intuitive understanding of how each component fit together. Some of this is down to space limits, as much of the mechanical details of algorithmic operation is sequestered to a large set of appendices. However, I think that it would improve the main text significantly to either have some concrete examples, or English-language "in other words..." parts to make the mathematical definitions (e.g., 4.1 - 4.3) and discussions of complexity & coverage less difficult to follow.



**Quality Of The Limitations Section:**

Limitations are addressed clearly

**Questions For Rebuttal:**

Mentioned in main review: Minor questions around the relationship of downward refinability to the concrete steps of their algorithm/definitions; a few questions around experimental analysis and the relationship between demonstration quality / task-type to success.

**Robotics Focus:**

Highly relevant to robotics but no hardware experiments

**Summary Of Paper:**

This paper is on bi-level planning in continuous-state/action spaces. Here, a high-level abstraction is used plan, which then guides low-level decisions. Existing bi-level planning methods often struggle in tasks where many objects in the environment change as a result of actions. This is because such planners must learn operator descriptions which track every single change between abstract states, and thus become unwieldy.

The insight of this paper is that operators need only describe the set of partial changes necessary to accomplish specific planning goals. To this end, they develop a formal procedure for distinguishing which changes are actually "necessary", and learn partial operators from a small batch of demonstrations via a hill-climbing search algorithm.

On experiments over the Behaviour-100 benchmark, along with a variety of additional tasks, the paper shows that their algorithm's success rate significantly exceeds many existing methods, including LOFT, CI, and GNNs.

**Summary Of Recommendation:**

This paper makes a solid technical contribution to bi-level planning, backed up by solid experiments. I am happy to recommend for acceptance. In the camera-ready, I would like to see some small tweaks to narrative clarity to make some of the mathematics easier to follow for the casual reader.

---

### Official Review · Reviewer_wErB · 2023-07-20

**Confidence:** 3
**Originality:** Good
**Technical Quality:** Excellent
**Clarity Of Presentation:** Excellent
**Impact:** 3

**Recommendation:**

Weak Accept: I recommend accepting the paper, but will not argue for my recommendation if the majority of other reviewers have a different opinion.

**Review:**

**Novelty:** As far as I know, the proposed objective for learning operators in terms of consistency with the given demonstration is novel.

**Clarity:** The aim and the overall idea are clearly explained, especially the introduction is good. However, the notations are a bit complex to understand.

**strengths**
1) The paper is very well written and has high-quality technical content. It is presented in a clear and easy-to-understand manner.
2) Learning operators from low-level demonstrations in a way that produces a feasible plan sketch for guiding low-level planning is indeed an interesting idea.
3) The video provided in the supplementary provides a very good overview of the approach.
4) The experiments are well-designed to address the specific questions under consideration.

**weakness**
1) No real robot experiments
2) The hill climbing search that is used to learn the operators may not scale for complex operators and environments.
3) The paper spends most time motivating and explaining the objective. However, a very compact explanation is provided on how to optimize the objective in the actual search performed.
4) At certain points, the introduction presents a general idea, yet the subsequent text lacks specific guidance on its implementation. For example,
    In section 4, the paper states, "To improve the coverage term in our objective, we wish to generate at least one new operator ω, and *potentially modify existing operators to cover this transition without uncovering any others*".  However, the text below only describes the process of learning a new operator to improve coverage, without providing an explanation of how or when existing operators can be modified to achieve this objective.
    In the introduction, the paper states, "we focus on the problem of learning operator descriptions from very few demonstrations given a set of predicates, an accurate low-level transition model, and *a set of parameterized controllers (such as Pick(x, y, z)) that serve as primitive actions.*" However, how to learn samplers for controllers is not included in the main text. Even though previous approaches have been adopted for this problem, stating it somewhere explicitly in the main text will help.



**Quality Of The Limitations Section:**

Limitations are addressed clearly

**Questions For Rebuttal:**

1) How scalable is this approach w.r.t to the complexity of the domain? Did you try some experiments, or do you have any plot that gives an idea about the learning time scales w.r.t the number of operators that needs to be learnt?


**Robotics Focus:**

Highly relevant to robotics but no hardware experiments

**Summary Of Paper:**

This paper focuses on learning symbolic operators in a manner that could be advantageous for low-level planning. Previous approaches learn operators that predict all the changes in state after the application of action.  The authors claim that learning to predict the changes that are relevant for high-level task planning will improve the overall planning performance. An objective function and a learning method are introduced for learning such a partial transition model. The validity of these claims is supported by conducting appropriate experiments.

**Summary Of Recommendation:**

The motivation and technical approach is well presented. The paper introduces a new way of approaching the problem of operator learning. In my opinion, theoretical contributions hold more significance than real robotic experiments for papers of this nature. Thus, the strengths of the paper surpass its weaknesses.

---

### Official Review · Reviewer_i5Gu · 2023-07-27

**Confidence:** 2
**Originality:** Very Good
**Technical Quality:** Very Good
**Clarity Of Presentation:** Fair
**Impact:** 4

**Recommendation:**

Strong Accept: I recommend accepting the paper and will argue for my recommendation even if other reviewers hold a different opinion.

**Review:**

Quality:
The work is well constructed. The authors present a well-structured and comprehensive study that addresses the challenging problem of learning symbolic operators for bilevel planning in complex robotics tasks. The research is backed by a sound theoretical foundation and is supported by empirical evaluations on various environments.

Clarity:
The main idea is well and explicitly introduced. However, for readers without extensive field knowledge, some parts may require frequent cross-referencing of the cited works. It would greatly enhance the paper's accessibility if the authors provided a more detailed background introduction, offering readers a better understanding of the context. Additionally, the scattered mathematical formulas throughout the paper can be challenging to follow, and it would greatly improve readability to consolidate all the symbols' meanings in a dedicated section.

Originality:
The IMPROVECOVERAGE generator is a novel algorithm that efficiently identifies abstract plan suffixes not covered by current operators and generates new ones to improve coverage. Additionally, the careful consideration of necessary changes within high-level states sets this work apart from conventional approaches.

Significance:
The significance of this work lies in its practical implications for solving long-horizon robotics problems with continuous state and action spaces. By learning symbolic operators that are better suited for high-level search and generalization, the approach enables efficient and effective bilevel planning in complex environments. The research contributes to advancing the field of task and motion planning, which is critical for real-world robotic applications.

Strengths:

Novel Objective: The focus on maximizing planning performance rather than prediction accuracy is a significant strength, as it aligns with the ultimate goal of efficient and effective planning in real-world scenarios.

IMPROVECOVERAGE Generator: The introduction of this algorithm is a major strength, as it efficiently identifies necessary changes and generates more general operators, leading to improved coverage and planning success.

Empirical Evaluations: The extensive experiments on various complex environments demonstrate the effectiveness and generalization capabilities of the proposed approach, adding credibility to the findings.

Weaknesses:

Data Requirements: The method relies on a set of predicates and demonstrations for operator learning. While it is effective with sufficient data, the approach's performance might be limited with sparse or insufficient data.

Computational Complexity: The IMPROVECOVERAGE generator, although effective, may suffer from computational inefficiency when the operator set becomes large or in more complex environments. Further optimization might be necessary.

**Quality Of The Limitations Section:**

Additional details required

**Questions For Rebuttal:**

Due to the reviwer's limited knowledge in the field, there are mainly experiment and implementation evaluation questions remain to be clarified:

Q1: Impact of Lambda Term: The authors mention the choice of the λ term in the objective function (Equation 1); however, they do not thoroughly discuss its impact on overall performance and sensitivity to different values. It would be helpful to include ablation studies concerning the λ parameter in Eq.1 to understand its effect better.

Q2: Complexity of IMPROVECOVERAGE: The computational expense of running the IMPROVECOVERAGE successor generator is not clearly addressed, and this could potentially lead to scalability issues in more complex domains. It would be beneficial to compare the computational cost of this work with related approaches to assess its efficiency.

Q3: Size of the Demonstration Dataset: Could the authors provide information on the scale of the demonstration dataset? Specifically, the number of trajectories or operators within the dataset. Additionally, it would be insightful to investigate how the performance of the bilevel planner is affected by varying amounts of demonstrations.

Q4: Details of Lower Level Controllers: The specifics of the lower-level controllers and their technical construction are not entirely clear. Further elaboration on this aspect is needed to better understand the transition model. Particularly, when performing planning at the abstract level, it would be crucial to know if the success of lower-level controllers is taken into consideration, as this could impact the overall planning process.


**Robotics Focus:**

Highly relevant to robotics but no hardware experiments

**Summary Of Paper:**

This work proposes a novel approach for learning symbolic operators to enhance bilevel planning in complex robotics tasks. The objective is to maximize the performance of the planning algorithm by focusing on necessary changes within high-level states. The key innovation, IMPROVECOVERAGE generator, efficiently identifies transitions not covered by current operators and generates new ones to improve overall coverage.  It leverages preimage backchaining to identify abstract plan suffixes within demonstrations that are not fully covered by the current set of operators, which aims to generate new operators and modify existing ones to cover specific transitions. The authors propose a novel method for constructing more general operators that cover multiple transitions in the dataset. By leveraging preimage backchaining and careful substitution, the operators are induced with appropriate preconditions and delete effects.

**Summary Of Recommendation:**

Based on the technical appendix and the main content of the work, it is evident that the authors have presented a comprehensive and well-structured study on bilevel planning algorithms. The technical appendix provides detailed descriptions of the algorithms, operators, samplers, and environments used in the experiments, which adds clarity and facilitates replication. The authors have also compared their proposed approach with several baselines, offering a thorough evaluation of its effectiveness. However, with limited knowledge in the field, some detailed aspects of the lower-level controller remain unclear. With the necessary revisions, this work has the potential to make a valuable contribution to the field of bilevel planning.

---

### Author Response · Authors · 2023-08-10
**Thank you for the thorough reviews!**

We thank the reviewers for their thorough and insightful suggestions and questions. We are very glad to hear that the reviewers felt that our paper
- tackles a challenging and important problem (Reviewers i5Gu, uCVm)
- presents an interesting and novel algorithmic solution (Reviewers i5Gu, wErB, sXyD, uCVm)
- presents well-designed, thorough experiments with convincing results (Reviewers i5Gu, wErB, sXyD)
- is generally well-written and clearly communicates the main ideas (Reviewers i5Gu, wErB, sXyD, uCVm)

Below, we first address a few shared questions. We then respond to reviewers' specific questions and concerns individually. We also include a revised version of our main paper attached to each rebuttal comment below, with changes and revisions highlighted (deletions are struck through in red).
1. *On the efficiency and scalability of our approach (Reviewers i5Gu and wErB)*: Our approach can be seen as trading off more computation compared to baselines (especially in the worst case) in order to learn simpler operators that generalize better to novel situations. In practice, we actually find that our approach's average wall-clock learning time is faster than all baselines across most of our domains (briefly mentioned in Section 7 of the main paper, with detailed experiment descriptions and results in Section G of our Appendix). We find that our method’s wall-clock time does increase and is worse than baselines on our two most complex BEHAVIOR tasks, but we find that this is because for certain random seeds, our approach learns extremely complex operators and thus the learning times for these seeds are significantly higher than for others, which inflates the average learning time. Additionally, while the current implementation does have a rather high worst-case time complexity, this could be significantly improved by implementing a better ImproveCoverage generator and using more sophisticated and efficient search algorithms than greedy hill-climbing. We have edited Section 7 slightly to better highlight these points.
1. *On the generalizability of planning with our learned operators (Reviewers sXyD, uCVm)*: The main source of generalization to new tasks for our approach comes from running bilevel planning with our learned operators. Specifically, we are able to solve novel tasks because our planner sequences together operators in novel combinations to accomplish previously unseen goals that might involve a different number of objects. As a simple example, consider the 'Cluttered 1D' domain where an agent moves along a 1D line strewn with objects that can be picked as long as the agent is close to that particular object. Suppose the agent is only ever shown demonstrations where its goal is to pick one particular object. If it learns good and sufficiently general 'Move' and 'Pick' operators to accomplish this, then planning with these operators will automatically generalize to goals where the agent must pick any number of objects (since any such goal can be accomplished by some combination of these learned operators). However, hyper-specific operators for 'Move' and 'Pick' will not chain together well to accomplish goals beyond those seen at training time. We empirically demonstrate this generalization in our experiments, since our evaluation tasks have "more objects, different initial states, and more atoms in the goal". It is true that this generalization is dependent on the learned operators, and thus on the precise demonstrations available at training time. A detailed characterization of how generalization is influenced by the the demonstrations, as well as the planning algorithm, is certainly an interesting direction for future work.

---

### Decision · Program_Chairs · 2023-08-30

**Decision:**

Accept (Poster)

**Comment:**

The paper proposes an improvement to bilevel planning with symbolic operators and neural samplers. The paper proposes a novel approach to learning operators that only model changes necessary for abstract planning to achieve specified goals. The paper is well written. In simulation experiments, the proposed approach outperforms baselines.